# Who Should Join the Decision-Making Table? Targeted Expert Selection for Enhanced Human-AI Collaboration

## Abstract

Integrating AI and human expertise can significantly enhance decision-making across various scenarios. This paper introduces a novel approach that leverages the Product of Experts (PoE) model to optimize decision-making by strategically combining AI with human inputs. While human experts bring diverse perspectives, their decisions may be constrained by biases or knowledge gaps. To address these limitations, we propose an AI agent that provides probabilistic, rule-based insights, complementing and filling human experts' knowledge gaps. A key feature of our approach is the strategic selection of human experts based on how well their knowledge complements or enhances the AI's recommendations. By dynamically adapting the expert selection process, we ensure that decisions benefit from the most impactful and complementary inputs. Our PoE model calibrates inputs from both AI and human experts, leveraging their combined strengths to improve decision outcomes. Furthermore, operating in an online setting, our framework can also continuously update the AI's knowledge and refine expert selection criteria, ensuring adaptability to evolving environments. Experiments in simulation environments demonstrate that our model effectively integrates logic rule-informed AI with human expertise, enhancing collaborative decision-making.

## 1 Introduction

In decision-making across various domains, human expertise is invaluable, but AI is increasingly being leveraged to augment and enhance these processes. Rather than replacing human specialists, as noted by Duan et al. (2019), a more promising approach is to combine AI with human knowledge, enhancing decision outcomes by utilizing the strengths of both. This collaboration between AI and human expertise has already proven beneficial in fields such as human resources (Davenport et al., 2010), marketing (Huang & Rust, 2022), and clinical radiology (Futoma et al., 2017; Bien et al., 2018), enabling more informed, comprehensive, and reliable decisions.

Consider the challenge of diagnosing and treating rare diseases in healthcare. Multidisciplinary treatment (MDT) is crucial in these cases, as it combines expertise across various fields to address the complexities of rare conditions. Despite this, individual biases and cognitive blind spots of human experts can still lead to suboptimal decisions. An AI doctor could play a critical role here by complementing human doctors' knowledge, filling gaps, and offering new perspectives based on its vast domain-specific data. This collaboration has the potential to improve diagnostic accuracy and treatment outcomes. Similarly, in academic peer review, bias and inconsistency can impact evaluations of research articles due to reviewers' subjective preferences and varied expertise. By integrating AI reviewers into this review process, we may bring objective, probabilistic insights to complement human judgment, improving fairness and efficiency in the review process.

Yet, key questions arise: *How should AI be effectively integrated into expert teams? How can AI and human decisions be combined to ensure optimal outcomes? And how can we ensure AI plays a positive, complementary role without introducing new risks or biases?* In this paper, we propose an AI-human collective decision-making framework to address these questions.

Our framework features a logic-informed AI agent designed to be transparent and interpretable. By grounding its decision-making process in rule-based logic, we aim to enhance human experts' trust in AI's recommendations and foster more effective collaboration with human experts.

At the core of our framework is the Product of Experts (PoE) model (Hinton, 2002; Cao & Fleet, 2014), which synthesizes AI and human inputs in a novel and elegant way. Unlike traditional ensemble methods such as bagging and boosting, PoE blends AI's probabilistic rule-based insights with the often more intuitive, experience-based judgments of human experts. This integration harmonizes diverse perspectives, ensuring that AI enhances decision-making without overriding human expertise. Additionally, a confusion matrix is used to assess and estimate the reliability of human contributors, enabling a more informed combination of AI and expert input.

Crucially, our framework incorporates an active perception mechanism for selecting human experts from a diverse pool, such as medical specialists within a hospital or across different institutions. This ensures that only the most informative and complementary expertise is utilized for each case. In our framework, the AI first generates initial recommendations, such as treatment options for a patient. Based on these recommendations, the system identifies and selects human experts whose insights complement and enhance the AI's output. Leveraging information gain, the framework automatically filters out less effective experts and adaptively chooses the most suitable ones for various scenarios and patients, such as those with specific rare conditions or complex needs, grouping them according to their strengths. The overall framework of the AI-human collaborative decision-making system operates in an online setting. As more patient data becomes available, our algorithm can further expand the AI's cognitive regions and refine its understanding of each expert's specialized areas, continuously improving decision-making, generalization, and predictive accuracy (Hoi et al., 2021). The architecture of our framework, illustrated in Fig. 1, highlights these innovations.

To summarize, our main contributions are as follows:

- We introduce a novel human-AI collaborative decision-making framework that integrates a logic-informed AI agent with the PoE model. This probabilistic model combines AI and human expertise, leveraging collective insights to enhance decision quality and reliability. The logic-informed AI agent ensures transparency and interpretability, grounding decisions in rule-based logic.

- Our framework features an active perception module that optimally selects human experts based on the estimation of their expertise. This module ensures that only the most relevant and impactful expertise is utilized, improving decision-making efficiency.

- We provide both theoretical analyses and empirical evidence to support the effectiveness of our framework. We demonstrate how the PoE model with the logic-informed AI agent and the active perception module improves decision accuracy and robustness. Additionally, we illustrate how the framework's adaptability to evolving data and environments is supported by empirical validation.

## 2 RELATED WORK

**Ensembles and Opinion Pools** Prior research has convincingly demonstrated the performance advantages of leveraging multiple predictors over a single predictor. This principle is evident in both model combinations (Kittler et al., 1998; Bagui, 2005; Sagi & Rokach, 2018) and human opinion aggregations (Hong & Page, 2004; Lamberson & Page, 2012). Majority voting (Dietterich, 2000) and naive Bayes aggregation (Xu et al., 1992) are prevalent methods for aggregating non-probabilistic classifiers. However, majority voting may fall short in accuracy enhancement with a limited number of predictors, and naive Bayes aggregation, while effective at the class level, does not fully exploit instance-level uncertainties presented by probabilistic labelers. In the realm of human opinion ensembling, methods range from additive linear and log-linear opinion pools for subjective distributions (Genest & Zidek, 1986), to techniques for weighting linear combinations of continuous human predictions (Davis-Stober et al., 2015), and voting strategies for consolidating label predictions from multiple human predictors (Lee & Lee, 2017). Our work differentiates itself by focusing on the integration of label-based human decisions with probabilistic model predictions, aiming to optimize the combination of these distinct sources of input.

**Human-AI Complementarity** Human-AI Complementarity aims to enhance the accuracy of predictions made by human experts utilizing decision support systems beyond the capabilities of

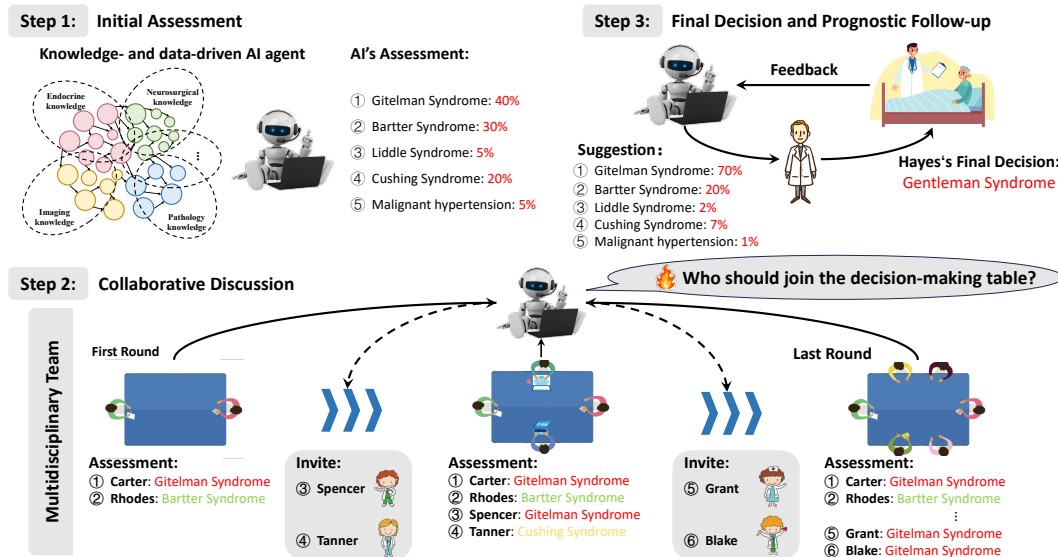

Figure 1: Overview of the proposed decision-making framework. Upon the arrival of a new patient, several doctors from different specialties and an AI agent engage in a collaborative discussion to provide their individual assessments. Subsequently, integrates these assessments, calibrating and combining them to produce a decision distribution. If the entropy of this distribution exceeds a specified threshold, and if fewer doctors than required have participated, additional experts are invited until the criteria are met. Following action implementation and based on the outcome, the encounter data is stored in a buffer for the AI agent to refine its decision-making model through rule learning.

the experts alone or the AI classifiers independently (De et al., 2019; De Toni et al., 2024). Despite this goal, empirical studies have frequently found that human-AI teams do not surpass the highest performance of either the human or the AI alone, even with AI explanations (Bansal et al., 2021; Liu et al., 2021). Several works model this challenge as a mixture of experts involving both humans and AI. This approach was initially introduced by (Madras et al., 2017) and later adapted by (Wilder et al., 2020) and (Pradier et al., 2021) with the introduction of a mixture of expert surrogates. However, these methods have often failed empirically due to difficulties in optimizing the loss function. Subsequent approaches have sought to improve these models, notably by enhancing calibration (Raman & Yee, 2021). Furthermore, (Buçinca et al., 2024) introduced offline reinforcement learning to develop decision support policies that optimize human-centric objectives, achieving improvements in joint human-AI accuracy. Nevertheless, these methods were not designed for contexts requiring collaboration between multiple humans and AI, thus overlooking the diversity in human groups. To address this gap, (Verma et al., 2023) introduced a model with ensemble prediction combining AI and human predictions, but optimization of collaboration costs is lacking. (Mozannar et al., 2023) formulated a novel surrogate loss function capable of deferring to one of the multiple users without combining AI and human predictions. Additionally, (Steyvers et al., 2022) used a Bayesian modeling framework to incorporate both human and machine predictions, demonstrating that hybrid human-machine models perform better than single models, but this approach overlooks model interpretability and human-AI interaction.

## 3    OUR PROPOSED AI-HUMAN COLLABORATIVE FRAMEWORK

In this section, we introduce our AI-human collaborative decision-making framework. We will use the multidisciplinary treatment (MDT) setting commonly encountered in healthcare as an example. In MDT, experts from diverse fields collaborate to diagnose and treat patients with complex medical conditions. This scenario naturally requires a collective approach that brings together varied expertise for optimal care.

**Problem Setup**    At each decision point, the system is presented with a patient characterized by features $x \in \mathcal{X}$, and it must select an action $a \in \mathcal{A}$ (e.g., a treatment option). The final decision on the action $a$ is made collectively by a human medical team with the support of an AI agent. The human

medical team consists of up to $L$ doctors, denoted as $H := \{h_1, \ldots, h_L\}$, each with potentially diverse areas of expertise. The AI agent, modeled as a probabilistic function $p(a \mid \boldsymbol{x}) \in \Delta^{|\mathcal{A}|}$, represents a probability distribution over possible actions. This probability simplex $\Delta^{|\mathcal{A}|}$ reflects the AI's assessment of the likelihood of different treatment options being successful, given the patient's features.

Once the medical team, with the aid of the AI agent, determines the final action, which will be executed, the system observes a reward $r \in \{0, 1\}$, indicating the patient's response to the chosen treatment. For example, the reward $r = 1$ reflects a successful outcome (e.g., recovery or improvement), while $r = 0$ reflects an unsuccessful outcome. We aim to design an AI-human collective decision-making system that effectively integrates human expertise with AI to make more efficient, informed, and superior decisions for patients.

In the following, we start by designing a **logic-informed AI agent** that uses domain knowledge, like logic rules, to ensure decisions are traceable and transparent, fostering trust and better collaboration. Next, we introduce the **PoE ensemble model**, which integrates probabilistic insights from the AI with human opinions, effectively combining diverse perspectives. We provide a simple theoretical justification to highlight the benefits of this probabilistic ensemble method. Then, we present the **active perception module** given the PoE model, which aims to sequentially select the most relevant human experts from the pool based on their expertise and prior decisions, ensuring weaker experts are filtered out of the decision-making process.

## 3.1 Design a Logic-Informed AI Agent

In healthcare, domain knowledge, such as detailed disease pathophysiology, treatment strategies, and clinical guidelines, is often represented as a compact set of logic rules. These rules can represent critical medical insights, including diagnostic criteria, therapeutic interventions, and best practices for managing complex or rare conditions. We will explore how to incorporate this rich medical knowledge as prior information to construct a logic-informed probabilistic AI agent. This AI agent, enhanced with domain-specific expertise, aims to bridge the knowledge gaps of human doctors.

We develop our logic-informed AI agent using the classic **Plackett-Luce model** (Maystre & Grossglauser, 2015), which is commonly used to model the probability of selecting an option from a set, based on the relative utility of each option. The core idea is to utilize the rule-based features to construct a utility function, which will then guide the AI agent in making probabilistic decisions.

**Logic Rules for Feature Construction** The domain knowledge is encoded as Horn rules, defining the *conditions under which particular actions should be taken*. For example, some of the Horn rule examples are as follows:

> Rule 1: If a patient shows specific symptoms, then a drug should be prescribed.
>
> Rule 2: If a patient previously responded positively to treatment, continue with the same treatment.
>
> Rule 3: If a patient is an adult with an underlying condition, consider surgery.

We will encode these rules into Boolean features that will be grounded from data to determine which action $a \in \mathcal{A}$, such as $a \in \{\text{Drug}, \text{Surgery}, \ldots\}$, to take. Specifically, for each action $a$, we construct Boolean features $\phi_a(\boldsymbol{x})$ based on the above described rules, such as

- For Drug Treatment (i.e., $a = \text{Drug}$), the logic-informed Boolean features include:

$$\phi_a(\boldsymbol{x}) = \begin{bmatrix} \mathbb{I}(\ \text{SymptomPresent}\ (\boldsymbol{x})) \\ \mathbb{I}(\ \text{ResponseToPreviousTreatment}\ (\boldsymbol{x})) \end{bmatrix}$$

- For Surgery (i.e., $a = \text{Surgery}$), the logic-informed Boolean features could be:

$$\phi_a(\boldsymbol{x}) = [\mathbb{I}(\text{Adult}(\boldsymbol{x}) \wedge \text{UnderlyingCondition}\ (\boldsymbol{x}))]$$

In these expressions, $\phi_a(\boldsymbol{x})$ are Boolean features reflecting whether specific conditions are met for action $a$. Intuitively, once the features associated with a particular action are grounded as true, the probability of selecting the corresponding action should be boosted. We will explicitly model this using the utility function.

**Utility Function Formulation**  The utility function for each action $a$ is expressed as a linear combination of the Boolean features derived from the logic rules:

$$\text{Utility}(a, \boldsymbol{x}) = \boldsymbol{w}_a^\top \boldsymbol{\phi}_a(\boldsymbol{x}) \tag{1}$$

where $\boldsymbol{w}_a$ represents the weight vector associated with action $a$, and $\boldsymbol{\phi}_a(\boldsymbol{x})$ is the Boolean feature vector specific to action $a$. This formulation allows the AI agent to compute a utility score for each action based on the observed features. To incorporate stochastic behavior into the action selection process, the selected action is the optimal solution of a random utility maximization problem:

$$\arg\max_a \text{Utility}(a, \boldsymbol{x}) + g_a \tag{2}$$

where $g_a$ denotes the independent Gumbel noise associated with action $a$. Given the model in Eq. (2), we can derive the AI-agent's probability of selecting action $a$ as follows:

$$p^{\text{AI}}(a \mid \boldsymbol{x}) = \frac{\exp\left(\text{Utility}(a, \boldsymbol{x})\right)}{\sum_{a' \in \mathcal{A}} \exp\left(\text{Utility}\left(a', \boldsymbol{x}\right)\right)}, \quad a \in \mathcal{A} \tag{3}$$

where $\text{Utility}(a, \boldsymbol{x})$ is modeled based on logic rules, as specified in Eq. (1). When using this AI agent in collaborative decision-making, we need to specify the rule weights, denoted as $\{\boldsymbol{w}_a\}_{a \in \mathcal{A}}$, and the set of logic rules, denoted as $\Gamma$, which have been estimated from historical data or calibrated from human expert input. In the next section, we will discuss how to estimate or refine these feature weights and the rule set, which is essential to making our model flexible, adaptable, and responsive to new data or expert knowledge.

## 3.2 Fusing Human Opinions and AI Recommendations

We propose the following Product-of-Experts model to integrate the probabilistic recommendations from the AI agent with the typically deterministic opinions made by human experts, combining them into a unified probabilistic framework. Suppose there is one AI agent $p^{\text{AI}}$ and $L$ human experts $\{h_1, \ldots, h_L\}$ have been invited, each possessing potentially different domain expertise. Given a feature $\boldsymbol{x}$, let $h_l(\boldsymbol{x})$ be the deterministic opinion of the $l$-th expert, and $p^{\text{AI}}(a \mid \boldsymbol{x})$ be the probabilistic recommendation provided by our logic-informed AI agent. Let $p_l(a \mid h_l(\boldsymbol{x}) = u_l)$ be the conditional probability that action $a$ is the true action given that the $l$-th expert's opinion is $u_l$. It transforms a deterministic opinion from human experts to a probabilistic recommendation. The true action, in this context, refers to the final action implemented.

Given human experts' opinions $u_1, \ldots, u_L$ and the AI agent's recommendation $p^{\text{AI}}(a \mid \boldsymbol{x})$, we propose to ensemble these opinions and recommendations using the following model:

$$p^{\text{PoE}}\left(a \mid \{h_l(\boldsymbol{x}) = u_l\}_{l=1,\ldots,L}, p^{\text{AI}}(a \mid \boldsymbol{x})\right) \propto \prod_{l=1}^{L} p_l(a \mid h_l(\boldsymbol{x}) = u_l) \cdot p^{\text{AI}}(a \mid \boldsymbol{x}), \quad a \in \mathcal{A}. \tag{4}$$

This model can be interpreted as a Product-of-Experts (PoE) model through augmenting the feature space. Indeed, consider a lifted parameter space $\Theta := \mathcal{X} \times \prod_{l=1}^{L} \{\phi_l(\boldsymbol{x}) : \boldsymbol{x} \in \mathcal{X}\}$, where for each $l = 1, \ldots, L$, when $h_l(\boldsymbol{x}) = u_l$, the feature mapping $\phi_l(\boldsymbol{x})$ maps $\boldsymbol{x}$ to a $|\mathcal{A}|$-dimensional one-hot vector $e_{u_l}$. On the parameter space $\Theta$, the formula (4) is a product-of-experts model with $L + 1$ experts, including $L$ human experts and one AI agent.

**Advantages of PoE in Collective Intelligence**  The PoE model offers an elegant approach to integrating AI and human expert opinions by multiplying and renormalizing their probability distributions. The PoE type of ensemble model provides several key appealing features:

1) *Reinforcing good actions*: In the region of features where all participants are highly accurate, i.e., the probabilities $p_l(a \mid h_l(\boldsymbol{x}) = u_l)$ and $p^{AI}(a \mid \boldsymbol{x})$ are closer to one, PoE model (4) exponentially amplifies the probability of the correct action, ensuring near-optimal decisions in those areas.

2) *Robust to influence from weak experts*: In regions of the feature space where not all experts are strong, the PoE model effectively balances their contributions. By preventing less confident experts from disproportionately influencing the outcome, the model ensures that the overall decision remains driven by the stronger experts. This stands in contrast to other ensemble models, where weak experts may dilute the decision. For a more detailed discussion, refer to Appendix C.

3) *Enabling a sequential expert selection with uncertainty quantification.* Thanks to its probabilistic nature, the PoE model enables us to identify and select the most informative and complementary experts, optimizing the decision-making process by minimizing decision entropy, as we will establish in Section 3.3.

**Human Expert Reliability Model** To build a probabilistic model $p(a \mid h_l(\boldsymbol{x})), \forall l \in \{1, \ldots, L\}$, for human experts given their deterministic opinions, we adopt a simple and effective expert reliability model, usually known as the confusion matrix, to learn the unknown cognitive processes. Formally, the confusion matrix of the $l$-th expert defines $\{p_l(k \mid h_l(\boldsymbol{x}) = u)\}_{k,u}$, which gives the probability that action $k$ is the true action when the $l$-th expert's deterministic opinion is $u$. We parameterize this relationship using a Softmax function and obtain the human expert reliability model:

$$p_l(k \mid h_l(\boldsymbol{x}) = u) = \frac{\exp\left(\psi_{k,u}^l\right)}{\sum_{k' \in \mathcal{A}} \exp\left(\psi_{k,u'}^l\right)}, \quad u, k \in \mathcal{A} \tag{5}$$

where the matrix $\psi^l := [\psi_{k,u}^l] \in R^{|\mathcal{A}| \times |\mathcal{A}|}$ are learned parameters, which can be interpreted as the confidence we place in expert $l$'s decision $u$ being correct when the optimal action is $k$. The estimation procedure of the confusion matrices of the $L$ experts is detailed in the next section.

**AI-Human Ensemble Model** Plugging the AI agent model (3) and the human expert reliability model (5) into the Product-of-Experts model (4), we arrive at our final AI-Human ensemble model

$$p^{\text{PoE}}\left(a \mid \{h_l(\boldsymbol{x}) = u_l\}_{l=1,\ldots,L}, p^{\text{AI}}(\boldsymbol{x})\right) = \frac{\exp\left(\sum_{l=1}^L \psi_{a,u_l}^l + \eta \operatorname{Utility}(a, \boldsymbol{x})\right)}{\sum_{a' \in \mathcal{A}} \exp\left(\sum_{l=1}^L \psi_{a',u_l}^l + \eta \operatorname{Utility}(a', \boldsymbol{x})\right)}, \quad a \in \mathcal{A} \tag{6}$$

It models the predicted probability of action $a$, given the feature $\boldsymbol{x}$ and the deterministic opinions of the human experts $\{u_l\}_{l=1,\ldots,L}$. The term $\sum_l \psi_{a,u_l}^l$ aggregates the influence of the experts' opinions, and the tuning hyper-parameter $\eta$ balances between human opinions and the AI agent's logic-rule-based utility.

In our AI-human collaborative system, it is important to note that the final decision rests with the human expert team, with the support of the AI agent in the decision-making process. If a human decision-maker feels that the current recommendation is insufficient for making a confident choice, the AI can identify and invite additional relevant experts from the pool to assist in the decision-making process. This adaptive mechanism ensures that human experts retain their ultimate authority throughout the decision-making process.

### 3.3 Targeted Expert Selection with Active Perception

Our AI-human PoE model has a natural advantage in quantifying uncertainty, which provides an additional benefit: targeted expert selection. For example, in a healthcare setting, this means that rather than consulting a large panel of experts—an approach that can be both inefficient and resource-intensive—we can strategically choose the experts whose insights are most pertinent to the patient's condition. This targeted approach not only reduces the time and cost associated with expert consultations but also enhances the accuracy and relevance of the final recommendations by focusing on the experts who provide the most valuable contributions.

We propose the following sequential expert selection scheme. The process begins with the AI agent making an initial decision by generating a conditional probability distribution over actions using its rule-based model (3). Based on this distribution, the system selects a human expert from the pool who is expected to provide the greatest information gain or reduce the current entropy of the decision model, i.e., the PoE model. Once the expert is consulted, the system updates the PoE distribution to incorporate their opinion. This process is repeated iteratively, with the system selecting the next expert from the remaining pool until either the entropy of the decision model, which measures the ambiguity of the recommendation, falls below a predefined threshold, or a specified number of experts have been consulted.

Specifically, let $H_l := \{h_1, h_2, \ldots, h_l\}$ represent the set of currently consulted experts with their opinions provided, and $H \setminus H_l$ be the set of remaining unconsulted experts. The goal at the current stage is to select the next expert $h_{l+1}$ from the remaining pool, aiming to maximize **information gain**, computed as

$$\max_{\tilde{h} \in H \setminus H_l} \mathbf{InfoGain}(\tilde{h}) := \mathcal{H}\left(p^{\text{PoE}}\left(\cdot \mid h_1(\boldsymbol{x}) = u_1, \ldots, h_l(\boldsymbol{x}) = u_l, p^{\text{AI}}(\boldsymbol{x})\right)\right)$$

$$- \mathcal{H}\left(\sum_{\tilde{u}} p_{\tilde{h}}(\tilde{u}) \cdot p^{\text{PoE}}\left(\cdot \mid h_1(\boldsymbol{x}) = u_1, \ldots, h_l(\boldsymbol{x}) = u_l, \tilde{h}(\boldsymbol{x}) = \tilde{u}, p^{\text{AI}}(\boldsymbol{x})\right)\right), \tag{7}$$

which is equivalent to maximizing the **reduction in decision entropy** (since the first term in the information gain is constant with respect to $\tilde{h}$). Therefore, the expert selection is expressed as:

$$h_{l+1}^*(\boldsymbol{x}) = \arg \min_{\tilde{h} \in H \setminus H_l} \mathcal{H}\left(\sum_{\tilde{u}} p_{\tilde{h}}(\tilde{u}) \cdot p^{\text{PoE}}\left(\cdot \mid \{h_i(\boldsymbol{x}) = u_i\}_{i=1,\dots,l}, \tilde{h}(\boldsymbol{x}) = \tilde{u}, p^{\text{AI}}(\boldsymbol{x})\right)\right) \quad (8)$$

where $\mathcal{H}(\cdot)$ represents the entropy of the current decision-making distribution, defined as $\mathcal{H}(p) = -\sum_{a \in \mathcal{A}} p(a) \log p(a)$, quantifying uncertainty in the distribution. A higher entropy indicates greater uncertainty, while a lower entropy reflects more confidence in specific outcomes. Here, $\tilde{u}$ denotes the potential opinions of the candidate expert. The prior $p(\tilde{u})$ can be estimated from historical data, capturing the likelihood of different opinions from the candidate expert. While developing a more precise model would require an estimator for $\hat{h}(\boldsymbol{x})$ based on features $\boldsymbol{x} \in \mathcal{X}$ for each expert, we simplify our approach in this paper by relying on the historical frequency of actions as the prior $p(\tilde{u})$.

This targeted expert selection approach ensures that the most informative and complementary experts will be chosen at each stage. The intuition is as follows. Let $p^{\text{PoE}}(\cdot \mid h_1(\boldsymbol{x}) = u_1, \dots, h_l(\boldsymbol{x}) = u_l, p^{\text{AI}}(\boldsymbol{x})) =: \mathbf{q}(\cdot)$ represent the probability distribution suggested by the current AI-human PoE model. The inclusion of an expert $\tilde{h} \in H \setminus H_l$ not in the current selection, with a confusion matrix $\tilde{\psi}$, would result in an updated probability distribution $\mathbf{q}_{\tilde{h}}(\cdot) := q_{\tilde{h}}(\cdot) \mathbf{q}(\cdot) / Z_{\tilde{h}}$, where $q_{\tilde{h}}(\cdot) := \sum_{\tilde{u}} p_{\tilde{h}}(\tilde{u}) \tilde{\psi}_{\tilde{u}}(\cdot)$ and $Z_{\tilde{h}}$ is a normalizing constant. Recall that due to the concavity of the function $q \mapsto -q \log q$, an entropy-minimizing distribution tends to approach a deterministic distribution. Consequently, if $-\mathbf{q}(\cdot) \log \mathbf{q}(\cdot)$ is relatively large for some actions $a$ and $a'$, it indicates that the current expert selection remains ambiguous between these two actions. To minimize the entropy (or equivalently maximize the information gain), our selection principle (8) is more likely to choose a complementary expert capable of distinguishing between actions $a$ and $a'$.

## 4 MODEL LEARNING

In this section, we discuss how to update the AI agent model (as defined in Eqs. (1) and (3)) and the human expert reliability model (as defined in Eq. (5)). By continuously refining these model parameters as data accumulates, we ensure our AI-human collaborative framework aligns with real-world needs, enhancing its accuracy and reliability in decision-making.

**Learning AI-Agent**  Given the observed data $(\boldsymbol{x}_i, a_i, r_i)$, $i = 1, \dots, n$, where $\boldsymbol{x}_i$ is the patient feature, $a_i$ is the executed action, and $r_i$ is the observed reward, we will learn or refine the logic rule set $\Gamma$ and their associated weights $\{\boldsymbol{w}_a\}_{a \in \mathcal{A}}$ by maximum likelihood estimation (MLE). However, the original MLE problem involves continuous updates of rule weights alongside a discrete rule set search, which is inherently a combinatorial problem. To tackle this complexity, we employ a column generation method, starting with a smaller, manageable problem that reduces the search space significantly. We iteratively identify new rules and update their weights, gradually expanding the search space as we advance. For a detailed description of the AI agent model learning algorithm, please refer to Appendix A.

**Learning Confusion Matrix of Human Experts**  We also address the update of the human expert reliability model, specifically the confusion matrices $\{\psi^l\}_{l=1,\dots,L}$. In scenarios with limited data, estimating the reliability of human agents (via confusion matrices) becomes challenging. To address this, we use a Bayesian approach and employ the Maximum A Posteriori (MAP) estimator for each parameter, assuming a Beta prior for the probability estimates. This allows for more robust estimates, even with small datasets. The detailed derivation and methodology are provided in the Appendix. B.

## 5 EXPERIMENTS

To assess the effectiveness of our proposed framework, we conducted a series of synthetic experiments. Our results demonstrate that the PoE model, optimized using an information gain objective, significantly outperforms other existing methods, including sparsely gated Mixture of Experts (MoE) (Shazeer et al., 2017), weighted voting, average voting, and various PoE model strategies within the current setting. Furthermore, we visualize the expert invitation process to illustrate how maximizing information gain can bridge cognitive gaps within a human-AI collaboration team. Finally,

we show that the performance of the AI agent improves with increased experience, ensuring both interpretability and accuracy as the agent learns over time.

**Experimental Setup** We generated synthetic datasets that emulate real-world conditions by incorporating a diverse array of rule-based simulation doctors and an AI agent. Initially, we predefined a set of ground truth rules, detailed in Table 2, to create the sample data. Each simulation doctor is characterized by distinct cognitive regions, allowing us to categorize them into three levels of expertise. Doctors 1, 2, and 3 specialize in actions 1, 2, and 3, respectively. They possess all ground truth rules relevant to their understanding regions but also include several erroneous rules. In contrast, Doctors 4, 5, and 6 demonstrate broader expertise, comprehending two actions and making mistakes in only one. Additionally, we introduced a random decision-maker, Doctor 7, who makes arbitrary decisions regardless of patient features. Our AI agent is relatively accurate, incorporating half of the ground truth rules that are consistent with our assumptions. It provides an initial treatment suggestion, which is then relayed to the responsible doctor, who assesses whether to invite additional doctors for their opinions. Following the treatment decision, the final choice is evaluated by our oracle, which possesses the complete set of ground truth rules and provides a reward based on Eq. 11. For a detailed description of the simulation process, please refer to Appendix Appendix D.

**Synthetic Data Generation** We divided the entire dataset into two disjoint subsets: (i) a training dataset $\mathcal{D}_t$ and (ii) an evaluation dataset $\mathcal{D}_e$. The training dataset $\mathcal{D}_t = \{\boldsymbol{x}_t, \{u_l\}_{l=1}^L, a_t, r_t, a^*\}$ includes comprehensive data such as consultation history and patient feedback, which are used to fit the calibration parameters. The evaluation dataset $\mathcal{D}_e$ is exclusively reserved for evaluation purposes. The optimal actions for each patient in all two datasets are initially derived from the rule set, followed by sampling-related symptoms, incorporating a degree of noise to simulate real-world uncertainties.

We designed two evaluation datasets with different levels of difficulty. In the Level-0 sample set, patients satisfy only one ground-truth rule within the corresponding optimal action and do not meet any other rules. In contrast, patients in the Level-1 sample set satisfy all ground-truth rules in their respective classes while also satisfying one ground-truth rule from another class. This setup ensures the reliability of our simulation experiments, as Level-1 patients are more suitable for Doctors 4, 5, and 6, whose expertise extends beyond the other doctors' domains.

**Evaluation Metrics** To assess the performance of our framework, we employed several key metrics, including accuracy and cumulative reward. Accuracy measures the likelihood that the final decisions made by different models align with the ground-truth labels. Cumulative reward quantifies the total reward accrued in the evaluation dataset, as determined by our oracle, a rule-based decision maker that utilizes the ground truth rule set.

## 5.1 Information Gain-Driven PoE Model Adaptation for Varied Contexts

We present a comprehensive comparison of our PoE model, aimed at maximizing information gain, as shown in Table. 1. Our framework is evaluated against traditional methods, along with an ablation study. Consistently, our approach outperforms all baseline models, highlighting its effectiveness in enhancing decision-making processes. Notably, when paired with a reasonably accurate AI agent, our method surpasses standalone models, even in scenarios where human models provide less reliable label suggestions. In contrast to ensemble techniques like MoE and traditional voting methods, our PoE model effectively filters out noisy or misleading information, leading to improved accuracy. A key concern with PoE models that lack a guiding AI agent (no AI) is their tendency to adapt to incorrect distributions, resulting in suboptimal decisions. We demonstrate the feasibility of training an interpretable and precise AI agent using established medical knowledge and expert insights. Our PoE model, when coupled with a calibrated AI agent, exhibits higher confidence than its uncalibrated version (no AI calibration). By applying temperature scaling, we can adjust the AI agent's probability distribution based on prior experience without altering the softmax function's peak, thereby controlling the entropy of the distribution, as shown in Guo et al. (2017). Compared to traditional PoE models that do not aim to maximize information gain, our approach proves more robust, particularly when experts have distinct cognitive regions.

Table 1: Comparison between different models in the synthetic experiments. The performance is evaluated by accuracy and cumulative rewards among 100 evaluation samples with 10 repeated experiments.

| Methods | Level-1 | | Level-0 | |
|---|---|---|---|---|
| | Accuracy ↑ | Rewards ↑ | Accuracy ↑ | Rewards ↑ |
| AI agent | $0.565 \pm 0.044$ | $85.8 \pm 3.86$ | $0.577 \pm 0.033$ | $71.3 \pm 3.22$ |
| Doctor1 | $0.399 \pm 0.021$ | $80.4 \pm 3.35$ | $0.385 \pm 0.043$ | $61.9 \pm 3.88$ |
| Doctor2 | $0.442 \pm 0.027$ | $81.9 \pm 2.98$ | $0.446 \pm 0.060$ | $63.1 \pm 3.85$ |
| Doctor3 | $0.459 \pm 0.049$ | $80.0 \pm 5.34$ | $0.415 \pm 0.028$ | $63.7 \pm 5.21$ |
| Doctor4 | $0.591 \pm 0.036$ | $87.1 \pm 2.58$ | $0.556 \pm 0.045$ | $68.6 \pm 2.83$ |
| Doctor5 | $0.596 \pm 0.026$ | $87.5 \pm 2.87$ | $0.559 \pm 0.043$ | $68.2 \pm 3.54$ |
| Doctor6 | $0.633 \pm 0.032$ | $87.8 \pm 2.90$ | $0.571 \pm 0.036$ | $68.0 \pm 4.47$ |
| Doctor7 | $0.267 \pm 0.024$ | $71.4 \pm 4.05$ | $0.280 \pm 0.039$ | $62.1 \pm 5.94$ |
| MoE | $0.523 \pm 0.051$ | $83.7 \pm 1.95$ | $0.476 \pm 0.045$ | $66.6 \pm 5.35$ |
| Majority voting | $0.325 \pm 0.009$ | $77.2 \pm 3.60$ | $0.330 \pm 0.013$ | $60.0 \pm 4.77$ |
| Weighted voting | $0.322 \pm 0.004$ | $80.7 \pm 4.36$ | $0.327 \pm 0.011$ | $60.3 \pm 3.28$ |
| PoE + Infogain | $\mathbf{0.659 \pm 0.042}$ | $\mathbf{89.0 \pm 1.94}$ | $\mathbf{0.668 \pm 0.051}$ | $\mathbf{72.3 \pm 4.02}$ |
| PoE + Infogain (no AI) | $0.428 \pm 0.053$ | $80.6 \pm 4.29$ | $0.371 \pm 0.051$ | $62.9 \pm 5.31$ |
| PoE + Infogain (no AI calibration) | $0.634 \pm 0.034$ | $87.1 \pm 2.70$ | $0.636 \pm 0.028$ | $71.3 \pm 3.97$ |
| PoE | $0.339 \pm 0.031$ | $76.3 \pm 5.22$ | $0.322 \pm 0.031$ | $58.1 \pm 5.56$ |
| PoE (no AI) | $0.337 \pm 0.031$ | $76.6 \pm 2.15$ | $0.340 \pm 0.052$ | $61.7 \pm 4.29$ |
| PoE (no AI calibration) | $0.338 \pm 0.030$ | $77.0 \pm 2.61$ | $0.330 \pm 0.043$ | $62.8 \pm 2.99$ |

## 5.2 VISUALIZATION OF THE SEQUENTIAL DECISION-MAKING PROCESS

To illustrate human-AI complementarity, we examine one of the most challenging samples to evaluate the performance of our algorithm. This sample satisfies all rules of its respective optimal action $a_1$ while also fulfilling one rule from each of the other classes. The features of this sample are defined as follows: $[x_0 = 1, x_1 = 1, x_2 = 0, x_3 = 1, x_4 = 1, x_5 = 0, x_6 = 1, x_7 = 0, x_8 = 1, x_9 = 1]$. Importantly, this instance meets the criteria for rules 1, 3, 4, and 5, which lie outside the expertise of both the AI agent and all other human models. We conduct this experiment to determine whether the PoE model can effectively capture the features of this complex sample by recovering the ground truth rule set and fine-tuning the output distribution to yield an accurate decision. As shown in Figure 2, we observe that inviting more informative human participants significantly enhances the recovery of the cognitive region of the human team. Furthermore, the entropy of the final distribution after calibration, illustrated in Figure 3, has been minimized, leading to a more accurate recovery of the ground truth conditional distribution.

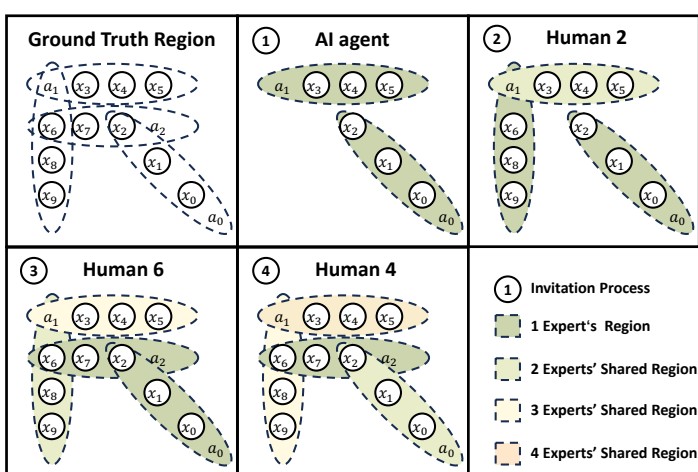

Figure 2: Dynamics of the cognitive region of human-AI team following sequential model invitation.

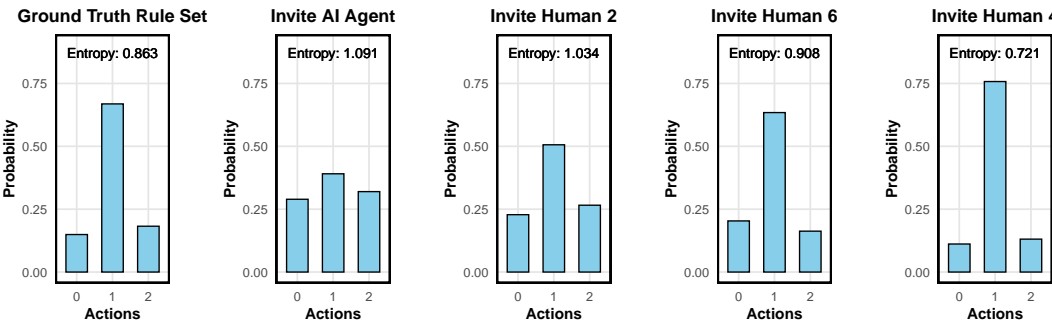

Figure 3: Comparison of probability distribution and entropy following sequential invitation.

## 5.3 VALIDATION OF THE AI AGENT'S COMPREHENSIVE COGNITIVE REGION

We validate the rule-learning capability of our AI agent by simulating real-world scenarios with a cohort of 20,000 pre-generated Level-0 patients. Initially, the AI agent provides its assessments and then strategically invites medical professionals based on the criterion of maximizing information gain. An oracle subsequently delivers feedback in the form of rewards, enabling us to update the AI agent's parameters every 5,000 patients. Detailed information about the experimental setup is provided in Appendix D. To assess the decision-making ability of our AI agent, we utilize accuracy and reward metrics, as illustrated in Figure 5. The mean absolute error (MAE) and standard deviation of the learned rule weights across ten replicates are reported in Table 5, alongside the accuracy of the learned rules in these replicates. Furthermore, we document the rules learned by the AI agent after each update in Table 6. The results conclusively demonstrate that our rule-learning method is highly effective in accurately identifying the ground-truth cognitive region with sufficient data. This validation shows that our AI agent can adapt and optimize its rule set as new data is incorporated, further supporting its application in complex, real-world decision-making environments.

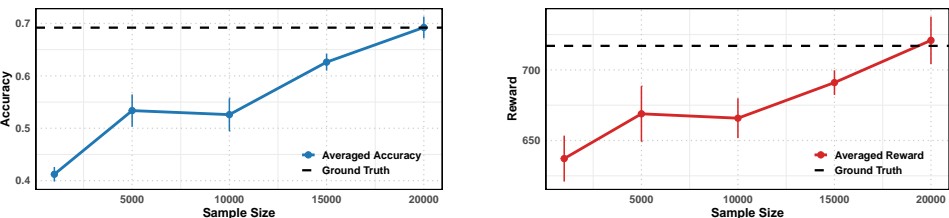

Figure 4: Learning curves for AI-agent on synthetic experiments. The x-axis represents the number of samples in the training dataset, while the y-axis shows the accuracy and the number of positive rewards in a total of 1000 evaluation samples.

## 6 CONCLUSION

In this paper, we proposed a novel AI-human collaborative decision-making framework using a Product of Experts (PoE) model to dynamically integrate AI and human expertise. Our approach strategically selects human experts based on their ability to complement the AI's rule-based insights, leading to more accurate and robust decisions. We provided both theoretical analysis and empirical validation of our method, demonstrating its superiority over traditional ensemble techniques and human-only approaches, particularly in complex, real-world-inspired scenarios. Additionally, we present a feasible mechanism for improving the AI agent online, allowing continuous adaptation to evolving environments. By ensuring transparency and interpretability through a logic-informed AI agent, our work pushes the boundaries of human-AI collaboration.

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

# A  LEARNING AI-AGENT

We consider a contextual bandit setting. At each round $t$, we receive a context $\boldsymbol{x}_t$ (i.e., features of a new patient). We choose an action $a_t \in \mathcal{A}$ (i.e., some type of treatment) and receive a reward $r_t \in \{0, 1\}$ (i.e., the survival or recovery condition of a patient). We assume

$$r(\boldsymbol{x}_t, a_t) \sim \text{Bernoulli}(f_{a_t}(\boldsymbol{x}_t)) \tag{9}$$

where for different actions $a \in \{1, \ldots, K\}$ we have distinct oracles $f_1(\boldsymbol{x}), \ldots, f_K(\boldsymbol{x})$. Given the history, $\{(\boldsymbol{x}_t, a_t, r_t)\}_{t=1,2,\ldots}$, these oracles are fit to the covariates and rewards from each arm separately. In other words,

$$\hat{f}_k = \arg\min_{f_k} \sum_{\{(\boldsymbol{x}_t, a_t, r_t) : a_t = k\}} \ell(r_t, f_k(\boldsymbol{x}_t)), \quad k = 1, \ldots, K, \tag{10}$$

where the loss can be the negative log-likelihood. Specifically, we make $f_k$ explainable and logic rule informed,

$$\log\left(\frac{p(r_t = 1 \mid \boldsymbol{x}_t, a_t = k)}{p(r_t = 0 \mid \boldsymbol{x}_t, a_t = k)}\right) = \log\left(\frac{f_k(\boldsymbol{x}_t)}{1 - f_k(\boldsymbol{x}_t)}\right) = \boldsymbol{w}_k^\top \boldsymbol{\phi}_k(\boldsymbol{x}_t) \tag{11}$$

Here, $\boldsymbol{w}_k = [w_{ki}] \in R^d$, $\boldsymbol{\phi}_k(\boldsymbol{x}_t) = [\phi_{ki}(\boldsymbol{x}_t)] \in \{0, 1\}^d$, and each feature $\phi_{ki}(\boldsymbol{x}_t)$ is the action-dependent rule-based binary feature specific $a_t = k$. We denote all the rule sets as $\Gamma = \{\Gamma_k\}$, where $\Gamma$ is the rule set and $\Gamma_k$ is the subset of rules associated with action type $k$.

At time round $t$, given the covariate $\boldsymbol{x}_t$, the AI agent makes the decision to maximize the patient's outcome according to

$$p(k = \arg\max_a r(\boldsymbol{x}_t, a)) = \frac{f_k(\boldsymbol{x}_t)}{\sum_{a \in \{1,\ldots,K\}} f_a(\boldsymbol{x}_t)} \tag{12}$$

It is more convenient to sample

$$a \sim \text{Mult}\left(\text{softmax}\left(\eta \times \text{sigmoid}^{-1}\left(f_1(\boldsymbol{x}_t), \ldots, f_k(\boldsymbol{x}_t)\right)\right)\right) \tag{13}$$

That is

$$p^{AI}(a = k \mid \boldsymbol{x}_t, \boldsymbol{w}, \Gamma) = \frac{\exp\left(\eta \boldsymbol{w}_k^\top \boldsymbol{\phi}_k(\boldsymbol{x})\right)}{\sum_{a \in \mathcal{A}} \exp\left(\eta \boldsymbol{w}_a^\top \boldsymbol{\phi}_a(\boldsymbol{x})\right)} \tag{14}$$

where $\eta$ trades off the exploration and exploitation. Increasing $\eta$ will drive the policy to do more exploitation. We treat $\eta$ as the calibrating parameter.

For the AI agent, the learning algorithm is as follows. The AI agent estimates $\hat{f}_1, \ldots, \hat{f}_K$ by fusing the information from prior knowledge and historical patient data. Denote the prior knowledge as $\Gamma^0 = \{\Gamma_k^0\}$, each rule weight and the additional rules distilled from data will be estimated by

$$\hat{\boldsymbol{w}}_k, \hat{\Gamma}_k = \arg\min_{\boldsymbol{w}, \Gamma \backslash \Gamma_k^0} \sum_{i \in \mathcal{D}: a_i = k} \ell(r_i, f_k(\boldsymbol{x}_i \mid \boldsymbol{w}, \Gamma)), \quad k = 1, \ldots, K \tag{15}$$

All the models $\hat{f}_1, \ldots, \hat{f}_K$ will be updated separately. For example, when we estimate $\hat{f}_k$, only the $\{(\boldsymbol{x}_t, a_t, r_t) : a_t = k\}$ will be used. We only explain how to estimate $\hat{f}_k$, and the same procedure can be applied to other models. We propose to update the rule set and each rule weight using the column generation type of algorithm. To make the derivation more convenient, we make some modifications and assume $\tilde{r}_t \in \{+1, -1\}$ and

$$p(r_t = 1 \mid \boldsymbol{x}_t, a_t = k) = \frac{\exp\left(\frac{1}{2}\boldsymbol{w}_k^\top \boldsymbol{\phi}_k(\boldsymbol{x}_t)\right)}{\exp\left(\frac{1}{2}\boldsymbol{w}_k^\top \boldsymbol{\phi}_k(\boldsymbol{x}_t)\right) + \exp\left(-\frac{1}{2}\boldsymbol{w}_k^\top \boldsymbol{\phi}_k(\boldsymbol{x}_t)\right)} \tag{16}$$

$$= \frac{1}{1 + \exp\left(-\boldsymbol{w}_k^\top \boldsymbol{\phi}_k(\boldsymbol{x}_t)\right)} \tag{17}$$

$$= \sigma(\boldsymbol{w}_k^\top \boldsymbol{\phi}_k(\boldsymbol{x}_t)), \tag{18}$$

$$p(r_t = -1 \mid \boldsymbol{x}_t, a_t = k) = \frac{\exp\left(-\frac{1}{2}\boldsymbol{w}_k^\top \boldsymbol{\phi}_k(\boldsymbol{x}_t)\right)}{\exp\left(\frac{1}{2}\boldsymbol{w}_k^\top \boldsymbol{\phi}_k(\boldsymbol{x}_t)\right) + \exp\left(-\frac{1}{2}\boldsymbol{w}_k^\top \boldsymbol{\phi}_k(\boldsymbol{x}_t)\right)} \tag{19}$$

$$= \frac{1}{1 + \exp\left(\boldsymbol{w}_k^\top \boldsymbol{\phi}_k(\boldsymbol{x}_t)\right)} \tag{20}$$

$$= \sigma(-\boldsymbol{w}_k^\top \boldsymbol{\phi}_k(\boldsymbol{x}_t)). \tag{21}$$

where $\sigma(\cdot)$ is the sigmoid function. The negative log-likelihood given the model assumption as described in Eq. (11) becomes

$$\ell(\boldsymbol{w}_k, \Gamma_k) = -\frac{1}{N} \sum_{n=1}^{N} \log \left( \sigma \left( \tilde{r}_n \boldsymbol{w}_k^\top \boldsymbol{\phi}_k(\boldsymbol{x}_n) \right) \right) \tag{22}$$

From now on, we will drop the subscript $k$. We formulate the overall model learning problem as an MLE problem, where the objective function is the negative log-likelihood, i.e.,

$$\text{Original Problem}: \boldsymbol{w}^*, \Gamma^* = \arg \min_{\boldsymbol{w}, \, \Gamma \backslash \Gamma^0} \ell(\boldsymbol{w}, \Gamma) \tag{23}$$

where $\ell(\boldsymbol{w}, \Gamma)$ is computed as Eq. (22). The above original problem is hard to solve, due to that the set of variables is exponentially large and can not be optimized simultaneously in a tractable way. We, therefore, use a divide-and-conquer idea and start with a small and durable problem, where the search space is much smaller, and we gradually increase the search space.

We start with $\Gamma^0$ and first learn the corresponding rule weights. Then, we expand this rule set to $\Gamma^1$ by solving a constructed subproblem for searching for a new rule to add. After that, we reestimate all the rule weights. The overall algorithm alternates between searching for a new rule and updating the rule weights, i.e., $\Gamma^0 \to \boldsymbol{w}_0 \to \Gamma^1 \to \boldsymbol{w}_1 \to \dots$. This will produce a nested sequence of subsets $\Gamma^0 \subset \Gamma^1 \subset \cdots \subset \Gamma^m \subset \cdots$.

Given the current candidate rule set $\Gamma^m$, the rule weights are updated by solving a (restricted) master problem formulated by

$$\text{RMP}: \quad \boldsymbol{w}_m^* = \arg \min_{\boldsymbol{w}} \ell(\boldsymbol{w}, \Gamma^m). \tag{24}$$

Let's further denote

$$y_m(\boldsymbol{x}) := \sum_{i=1}^{m} w_i \phi(\boldsymbol{x}; \gamma_i) \tag{25}$$

where $y_m(\boldsymbol{x})$ is constructed by $m$ adaptive basis functions, and for each basis, $w_i \phi(\boldsymbol{x}; \gamma_i)$, we let $w_i$ be the rule weight and $\gamma_i$ encode the rule content.

In other words, after the rule weights update, we have

$$\forall \gamma_i \in \Gamma^m \quad \frac{\partial \ell}{\partial w_i} = 0 \tag{26}$$

We will construct a subproblem to do the rule search and get $\Gamma^{m+1}$ from $\Gamma^m$.

Given $\Gamma^m$ and $\boldsymbol{w}_m^* := [w_{1:m}^*]$, where each element $w_i^* \neq 0$, we construct a new rule weight vector by augmenting one extra dimension, $\boldsymbol{w}_{m+1} := [w_{1:m}^*, w_{m+1}]$, where $w_{m+1} = 0$. The new rule is determined by computing the gradient with respect to the weights from the remaining rule set,

$$\forall \gamma_i \in \Gamma \backslash \Gamma^m, \quad \frac{\partial \ell}{\partial w_i} = \frac{\partial \ell}{\partial y_{m+1}} \frac{\partial y_{m+1}}{\partial w_i} = \frac{1}{N} \sum_{n=1}^{N} \frac{-\tilde{r}_n \exp\left(-\tilde{r}_n \boldsymbol{w}_m^{*\top} \boldsymbol{\phi}_m(\boldsymbol{x}_n)\right)}{1 + \exp\left(-\tilde{r}_n \boldsymbol{w}_m^{*\top} \boldsymbol{\phi}_m(\boldsymbol{x}_n)\right)} \cdot \phi_{m+1}(\boldsymbol{x}_n) \tag{27}$$

A subproblem is formulated to propose a new logic rule, which can potentially best decrease the loss function.

Suppose we force all the rule weights to be positive, then the subproblem to find the rule that yields the most negative gradient, i.e.,

$$\text{Subproblem}: \min_{\gamma_i} \frac{\partial \ell}{\partial w_i}, \quad \forall \gamma_i \in \Gamma \backslash \Gamma^m \tag{28}$$

Suppose we allow the rule weight can be both positive and negative, we search for a new rule so that the *magnitude* of the gradient is maximized, i.e.,

$$\text{Subproblem}: \max_{\gamma_i} \left| \frac{\partial \ell}{\partial w_i} \right|, \quad \forall \gamma_i \in \Gamma \backslash \Gamma^m \tag{29}$$

Intuitively, since the new rule hasn't been added and its corresponding rule weight is zero at the moment if the weight gradient is negative, we must increase the rule weight (i.e., the new weight may be positive) to decrease the current loss function. If the gradient is positive, we need to decrease the rule weight (i.e., the new weight might be negative) to decrease the current loss function. The new rule to add is the one that yields the maximal magnitude of the gradient.

**Stopping rule**   If we continue to run the above algorithm, the algorithm will not terminate until every possible rule has been added to the rule, i.e., the model contains no non-zero weights. In this case, there is no meaning to using the above column generation type of algorithm. In practice, we will set the stopping threshold, $\lambda$, and will terminate the algorithm if the maximal magnitude of the gradient does not exceed $\lambda$, i.e.,

$$\max_{\gamma_i} \left| \frac{\partial \ell}{\partial w_i} \right| < \lambda \tag{30}$$

Here, $\lambda$ is the hyperparameter and tradeoff the model flexibility and sparsity.

**AI-agent Calibration**   We treat $\eta$ as the calibrating parameter determined by the host. We will adopt the Platt scaling (temperature scaling) Platt et al. (1999) Niculescu-Mizil & Caruana (2005) for calibration.

The main idea is we can use the historical data $\mathcal{D}$ to estimate

$$\hat{\boldsymbol{w}}_k, \hat{\Gamma}_k = \arg \min_{\boldsymbol{w}, \Gamma} \sum_{i \in \mathcal{D}: a_i = k} \ell(r_i, f_k(\boldsymbol{x}_i \mid \boldsymbol{w}, \mathcal{F})), \quad k = 1, \ldots, K \tag{31}$$

The primary goal is to minimize the negative log-likelihood. Notably, $\eta$ adjusts the entropy of the output distribution without altering the location of the maximum in the softmax function output, thereby finely tuning the probability distribution across various classes.

---

**Algorithm 1** Optimized Learning Process of AI agent

---

**Initialization:** Initialize Oracles $\hat{f}_{1:K}$, Data Buffer $\mathcal{D}_{1:K}$, Calibration Parameters $\eta$ and $\psi$.

1: **for** each patient $t = 1, 2, \ldots$ **do**
2:    **Receive** context $\boldsymbol{x}_t$.
3:    **Update** AI agent calibration parameters $\eta$ and estimate parameter $\psi_{k,u}^l$ for each expert reliability model.
4:    **if** information gain threshold is satisfied **then**
5:       **Sample** action $a_t$ according to the policy:

$$a_t = \arg \max_k p^{\text{PoE}}(k \mid h_1(\boldsymbol{x}) = u_1, \ldots, h_l(\boldsymbol{x}) = u_l, p^{\text{AI}}(\boldsymbol{x}))$$

6:    **else**
7:       **Minimize** the entropy of the policy function and invite corresponding experts with Eq. 8:

$$h_{l+1}^*(\boldsymbol{x}) = \arg \min_{\tilde{h} \in H \setminus H_l} \mathcal{H} \left( \sum_{\tilde{u}} p_{\tilde{h}}(\tilde{u}) \cdot p^{\text{PoE}} \left( \cdot \mid \{h_i(\boldsymbol{x}) = u_i\}_{i=1,\ldots,l}, \tilde{h}(\boldsymbol{x}) = \tilde{u}, p^{\text{AI}}(\boldsymbol{x}) \right) \right)$$

8:    **Sample** action $a_t$ according to the new policy:

$$a_t = \arg \max_k p^{\text{PoE}}(k \mid h_1(\boldsymbol{x}) = u_1, \ldots, h_L(\boldsymbol{x}) = u_l, h_{l+1}^*(\boldsymbol{x}) = u_{l+1}, p^{\text{AI}}(\boldsymbol{x}))$$

9:    **Observe** reward $r_t$ and **store** $(\boldsymbol{x}_t, a_t, r_t)$ in $\mathcal{D}_{a_t}$.
10:   **Update** the oracle for the action taken:

$$\hat{\boldsymbol{w}}_k, \hat{\Gamma}_k = \arg \min_{\boldsymbol{w}, \Gamma} \sum_{i \in \mathcal{D}: a_i = k} \ell(r_i, f_k(\boldsymbol{x}_i \mid \boldsymbol{w}, \mathcal{F})), \quad k = 1, \ldots, K$$

---

## B   LEARNING THE CONFUSION MATRIX

To address the small data challenge, we employ a Bayesian approach and leverage the Maximum A Posteriori (MAP) estimator for the human expert reliability model. Specifically, we focus on a probabilistic model $p(a \mid h_l(\boldsymbol{x}))$ for each human expert $l$. The confusion matrix $\psi^l = [\psi_{k,u}^l] \in \mathbb{R}^{|\mathcal{A}| \times |\mathcal{A}|}$ encodes the confidence we place in expert $l$'s decision $u$ being correct when the optimal action is $k$. This relationship is modeled as:

$$\log \left( \frac{p(r_t = 1 \mid a_t = k, h_l(\boldsymbol{x}_t) = u)}{p(r_t = 0 \mid a_t = k, h_l(\boldsymbol{x}_t) = u)} \right) = \psi_{k,u}^l \tag{32}$$

From this formulation, we can recover the probability as follows:

$$p(r_t = 1 \mid a_t = k, h_l(\boldsymbol{x}_t) = u) = \frac{1}{1 + \exp(-\psi_{k,u}^l)} \tag{33}$$

For notational simplicity, we define:

$$p(r_t = 1 \mid a_t = k, h_l(\boldsymbol{x}_t) = u) := p_{k,u}^l$$

Given the observed data $\mathcal{D} = \{h_l(\boldsymbol{x}_t), a_t, r_t\}_{t=1}^T$, where $h_l(\boldsymbol{x}_t)$ represents the action recommended by human expert $l$, $a_t$ is the executed action, and $r_t$ is the observed reward, we can estimate the confusion matrix $\psi^l = [\psi_{k,u}^l] \in \mathbb{R}^{|\mathcal{A}| \times |\mathcal{A}|}$ using Maximum Likelihood Estimation (MLE), based on instances where $a_t = k$ and expert $l$'s recommendation was $u$.

Considering the small data regime, we adopt a Bayesian approach and aim to estimate the MAP value $\hat{p}_{k,u}^{\text{MAP}}$. We assume a Dirichlet prior for $p_{k,u}$, which reduces to a Beta distribution in the binary case:

$$p_{k,u} \mid \alpha, \beta \sim \text{Beta}(\alpha, \beta) \tag{34}$$

$$p(p_{k,u} \mid \alpha, \beta) \propto p_{k,u}^{\alpha-1} (1 - p_{k,u})^{\beta-1} \tag{35}$$

The posterior distribution, combining the likelihood with the prior, becomes:

$$p_{k,u} \mid \mathcal{D}, \alpha, \beta \sim \text{Beta} \left( \alpha + \sum_t r_t, \beta + \sum_t (1 - r_t) \right) \tag{36}$$

Thus, the MAP estimate for $p_{k,u}$ is given by:

$$\hat{p}_{k,u}^{\text{MAP}} = \frac{\alpha - 1 + \sum_{t:a_t=k, h_l(\boldsymbol{x}_t)=u} r_t}{\alpha + \beta - 2 + n_{k,u}^l} \tag{37}$$

where $n_{k,u}^l$ represents the number of instances where action $h_l(\boldsymbol{x}_t) = u$ was recommended, and the true action $a_t = k$ was implemented. This formulation is analogous to the standard Beta-Bernoulli posterior.

## C  ADVANTAGES OF THE PROPOSED PoE MODEL IN COLLECTIVE INTELLIGENCE

Let $\mathcal{R}_l \subseteq \mathcal{X}$ be the specialized region of Expert $l$, $l = 1, \ldots, L$. In the specialized region $\mathcal{R}_l$, the $l$-th expert has a high prediction precision. The conditional probability of $a$ is the optimal action given the $l$-th expert's opinion is $a$, is represented as

$$p_l(a \mid h_l(\boldsymbol{x}) = a) = \begin{cases} 1 - \epsilon_l, & \text{if } a = a^* \\ \epsilon_l, & \text{if } a \neq a^* \end{cases}$$

where $\epsilon_l > 0$ is a small number.

## C.1 PoE vs. MoE in Common Regions

On the specialized region $\mathcal{R}_1 \cap \mathcal{R}_2 \cap \cdots \cap \mathcal{R}_L$, all experts will choose the optimal action $a$ with high probability.

To simply notation, we denote

$$p^{\text{PoE}}(a \mid \boldsymbol{x}) := p^{\text{PoE}}(a \mid h_1(\boldsymbol{x}) = \cdots = h_L(\boldsymbol{x}) = a).$$

For any $a' \neq a$, it holds that

$$\frac{p^{\text{PoE}}(a \mid \boldsymbol{x})}{p^{\text{PoE}}(a' \mid \boldsymbol{x})} = \prod_{l=1}^{L} \left( \frac{1 - \epsilon_l}{\epsilon_l} \right) \cdot \frac{1 - \epsilon_{\text{AI}}}{\epsilon_{\text{AI}}}.$$

Suppose $\epsilon_l = \epsilon < 1/2$, then we have

$$\frac{p^{\text{PoE}}(a \mid \boldsymbol{x})}{p^{\text{PoE}}(a' \mid \boldsymbol{x})} = \left( \frac{1 - \epsilon}{\epsilon} \right)^L \frac{1 - \epsilon_{\text{AI}}}{\epsilon_{\text{AI}}} \tag{38}$$

increases exponentially with the number of experts $L$, which boosts the prediction precision exponentially.

For the MoE Model, we denote

$$p^{\text{MoE}}(a \mid \boldsymbol{x}) := p^{\text{MoE}}(a \mid h_1(\boldsymbol{x}) = \cdots = h_L(\boldsymbol{x}) = a).$$

The MoE model takes an average over probabilities, which results in

$$\frac{p^{\text{MoE}}(a \mid \boldsymbol{x})}{p^{\text{MoE}}(a' \mid \boldsymbol{x})} = \frac{\frac{1}{L} \sum_{l=1}^{L} (1 - \epsilon_l) + \eta p^{\text{AI}}(a \mid \boldsymbol{x})/L}{\frac{1}{L} \sum_{l=1}^{L} \epsilon_l + \eta p^{\text{AI}}(a' \mid \boldsymbol{x})/L}.$$

Assuming $\epsilon_l = \epsilon$ for all experts, then the ratio

$$\frac{p^{\text{MoE}}(a \mid \boldsymbol{x})}{p^{\text{MoE}}(a' \mid \boldsymbol{x})} = \frac{(1 - \epsilon) + \eta p^{\text{AI}}(a \mid \boldsymbol{x})}{\epsilon + \eta p^{\text{AI}}(a' \mid \boldsymbol{x})} \tag{39}$$

does not scale with $L$ and thus the common opinion is not reinforced.

Comparing (39) and (38), we see that, on a common region, the PoE model benefits significantly more from the multiplicative combination of experts' opinions.

## C.2 PoE vs. MoE in Uncommon Regions

In uncommon regions, the probability $p_l(a \mid h_l(\boldsymbol{x}_t))$ of some expert $l$ may be spread out across non-optimal actions in $\mathcal{A}$. An expert is considered weak in a specific region if the action with the highest probability is incorrect. Let $E_w$ denote the set of weak experts and $E_s$ denote the set of strong experts.

Using the notation from the previous case, we have that

$$\frac{p^{\text{PoE}}(a \mid \boldsymbol{x})}{p^{\text{PoE}}(a' \mid \boldsymbol{x})} = \prod_{l \in E_s} \left( \frac{1 - \epsilon_l}{\epsilon_l} \right) \prod_{l \in E_w} \frac{p_l(a \mid h_l(\boldsymbol{x}) = u_l)}{p_l(a' \mid h_l(\boldsymbol{x}) = u_l)}.$$

Outside their specialized region, the experts' opinions are less accurate. Supposed they are close to random guesses:

$$p_l(a \mid h_l(\boldsymbol{x}) = u_l) \approx \frac{1}{|\mathcal{A}|}, \quad \forall a, u \in \mathcal{A}, \boldsymbol{x} \notin \mathcal{R}_l.$$

Assuming $\epsilon_l = \epsilon$ for all experts, then the above ratio becomes

$$\frac{p^{\text{PoE}}(a \mid \boldsymbol{x})}{p^{\text{PoE}}(a' \mid \boldsymbol{x})} \approx \left( \frac{1 - \epsilon}{\epsilon} \right)^{|E_s|}. \tag{40}$$

This shows that the prediction of the PoE model is to deteriorate too much due to the existence of weak experts.

In contrast, for a MoE model, we have

$$\frac{p^{\text{MoE}}(a \mid \boldsymbol{x})}{p^{\text{MoE}}(a' \mid \boldsymbol{x})} = \frac{\frac{1}{L}\sum_{l \in E_s}(1-\epsilon_l) + \frac{1}{L}\sum_{l \in E_w} p_l(a \mid h_l(\boldsymbol{x})=u_l) + \eta p^{\text{AI}}(a \mid h_l(\boldsymbol{x})=u_l)/L}{\frac{1}{L}\sum_{l \in E_s}\epsilon_l + \frac{1}{L}\sum_{l \in E_w} p_l(a \mid h_l(\boldsymbol{x})=u_l) + \eta p^{\text{AI}}(a' \mid h_l(\boldsymbol{x})=u_l)/L}.$$

Then under the situation above, we have

$$\frac{p^{\text{MoE}}(a \mid \boldsymbol{x})}{p^{\text{MoE}}(a' \mid \boldsymbol{x})} = \frac{(1-\epsilon)|E_s| + |E_w|/|\mathcal{A}| + \eta p^{\text{AI}}(a \mid h_l(\boldsymbol{x})=u_l)}{\epsilon|E_s| + |E_w|/|\mathcal{A}| + \eta p^{\text{AI}}(a' \mid h_l(\boldsymbol{x})=u_l)}.$$

Compared this with equation 40, in the MoE model, weak experts can have a larger influence on the final decision because their opinions are averaged with the strong experts, diluting the strength of the more confident predictions.

# D    SETTING OF SYNTHETIC EXPERIMENTS

The framework integrates three principal entities: an AI host, an AI agent, and human agents. We assess their performance through a series of simulations, employing three distinct sets of synthetic data, each derived from a unique ground truth rule set. Below, we detail the architecture of our simulation framework.

**Patient Simulator**    The sample generation process is rooted in a predefined set of ground truth rules, as shown in Table. 2. Initially, the generator selects a set of actions based on the rule weights, simulating population-level decision-making processes. This selection is formalized by the equation:

$$a \sim \text{Mult}\left(\text{softmax}\left(\text{sigmoid}^{-1}\left(f_1(\boldsymbol{x}_t), \ldots, f_k(\boldsymbol{x}_t)\right)\right)\right),$$

where $f(\boldsymbol{x}_t)$ represents the feature functions associated with the rule set, and the actions are selected if the corresponding features satisfy the rule conditions.

After action selection, we initially generate a random binary sequence. For a Level-0 patient, at least one of the rules corresponding to the selected actions must be satisfied, while none of the rules associated with other actions should be met. For a Level-1 patient, all two rules from the selected actions must be satisfied, along with at least one rule from the other actions. This definition ensures varying levels of difficulty tailored to the expertise of different decision-makers, reflecting their respective cognitive domains.

Table 2: The rules assigned to each rule-based decision-maker.

| Head Predicate | Rules | Weight |
|---|---|---|
| $a_0$ | 1: $x_0 \wedge x_1 \wedge \neg x_2$ | 1.5 |
| | 2: $x_3 \wedge x_7 \wedge \neg x_9$ | 1.5 |
| $a_1$ | 3: $x_3 \wedge x_4 \wedge \neg x_5$ | 1.4 |
| | 4: $x_6 \wedge x_7 \wedge x_9$ | 1.6 |
| $a_2$ | 5: $\neg x_2 \wedge x_6 \wedge \neg x_7$ | 1.7 |
| | 6 : $x_5 \wedge \neg x_8 \wedge x_9$ | 1.4 |

**Human Doctor Simulator**    In real-world healthcare environments, physicians from various specialties exhibit distinct domains of expertise and may even hold inaccurate beliefs. To simulate this, we model each human doctor as a rule-based probabilistic decision-maker endowed with a unique set of rules that reflect their specific expertise and biases. These rule sets can demonstrate preferences for specific treatments or deviate significantly from the established ground-truth rule set. Unlike real-world scenarios where doctors typically provide deterministic treatment choices, our simulated doctors select the treatment corresponding to the probability distribution from the softmax function.

Our simulation framework includes a diverse pool of seven doctors. For each patient scenario, the AI agent first offers a treatment suggestion. If the entropy of the collective treatment distribution does not meet a predefined threshold (set at 0.3 in all our experiments), indicating a lack of consensus or insufficient confidence among the initial doctors, the AI host intervenes. The AI host then invites additional doctors from the pool to contribute their recommendations, aiming to refine the decision-making process and enhance the reliability of the treatment choice. Details of all eight rule-based decision-makers can be found in Table 3.

Table 3: The rules assigned to each rule-based decision-maker.

| Model | Rule Set | Weight | Model | Rule Set | Weight |
|---|---|---|---|---|---|
| AI Agent | $a_0 \leftarrow x_0 \wedge x_1 \neg x_2$ | 1.4 | Human 1 | $a_0 \leftarrow x_0 \wedge x_1 \wedge \neg x_2$ | 1.5 |
| | $a_0 \leftarrow x_3 \wedge x_7 \wedge \neg x_9 \wedge x_8$ | 1.6 | | $a_0 \leftarrow x_3 \wedge x_7 \wedge \neg x_9$ | 1.2 |
| | $a_1 \leftarrow x_3 \wedge x_4 \wedge \neg x_5$ | 1.7 | | $a_1 \leftarrow x_3 \wedge \neg x_4$ | 1.3 |
| | $a_1 \leftarrow x_6 \wedge x_7 \wedge x_8 \wedge x_9$ | 1.3 | | $a_1 \leftarrow x_6 \wedge x_8 \wedge \neg x_9$ | 1.4 |
| | $a_2 \leftarrow \neg x_2 \wedge x_6 \wedge \neg x_7$ | 1.5 | | $a_2 \leftarrow \neg x_2 \wedge x_6 \wedge x_7 \wedge \neg x_8$ | 1.1 |
| | $a_2 \leftarrow x_5 \wedge \neg x_8 \wedge x_9 \wedge x_3$ | 1.5 | | $a_2 \leftarrow x_5 \wedge \neg x_9$ | 1.6 |
| Human 2 | $a_0 \leftarrow x_0 \wedge \neg x_1$ | 1.5 | Human 3 | $a_0 \leftarrow x_0 \wedge \neg x_1$ | 1.5 |
| | $a_0 \leftarrow x_2 \wedge x_3 \wedge x_7 \wedge \neg x_9$ | 1.2 | | $a_0 \leftarrow x_2 \wedge x_3 \wedge \neg x_9$ | 1.2 |
| | $a_1 \leftarrow x_3 \wedge x_4 \wedge \neg x_5$ | 1.3 | | $a_1 \leftarrow x_3 \wedge x_4 \wedge x_6$ | 1.3 |
| | $a_1 \leftarrow x_6 \wedge x_8 \wedge x_9$ | 1.4 | | $a_1 \leftarrow x_6 \wedge \neg x_8$ | 1.4 |
| | $a_2 \leftarrow \neg x_2 \wedge x_6 \wedge x_7 \wedge \neg x_8$ | 1.1 | | $a_2 \leftarrow \neg x_2 \wedge x_6 \wedge \neg x_7$ | 1.1 |
| | $a_2 \leftarrow x_5 \wedge \neg x_9$ | 1.6 | | $a_2 \leftarrow x_5 \wedge \neg x_8 \wedge x_9$ | 1.6 |
| Human 4 | $a_0 \leftarrow x_0 \wedge x_1 \wedge \neg x_2$ | 1.5 | Human 5 | $a_0 \leftarrow x_0 \wedge x_1 \wedge \neg x_2$ | 1.5 |
| | $a_0 \leftarrow x_3 \wedge x_7 \wedge \neg x_9$ | 1.4 | | $a_0 \leftarrow x_3 \wedge x_7 \wedge \neg x_9$ | 1.5 |
| | $a_1 \leftarrow x_3 \wedge x_4 \wedge \neg x_5$ | 1.6 | | $a_1 \leftarrow x_3 \wedge \neg x_4$ | 1.7 |
| | $a_1 \leftarrow x_6 \wedge x_8 \wedge x_9$ | 1.4 | | $a_1 \leftarrow x_5 \wedge x_6 \wedge x_8 \wedge x_9$ | 1.3 |
| | $a_2 \leftarrow \neg x_2 \wedge \neg x_6$ | 1.4 | | $a_2 \leftarrow \neg x_2 \wedge x_6 \wedge \neg x_7$ | 1.6 |
| | $a_2 \leftarrow x_5 \wedge x_7 \wedge \neg x_8 \wedge x_9$ | 1.6 | | $a_2 \leftarrow x_5 \wedge \neg x_8 \wedge x_9$ | 1.4 |
| Human 6 | $a_0 \leftarrow x_0 \wedge \neg x_1$ | 1.5 | Human 7 | $a_0 \leftarrow x_0 \wedge \neg x_1$ | 1.5 |
| | $a_0 \leftarrow x_2 \wedge x_3 \wedge x_7 \wedge \neg x_9$ | 1.4 | | $a_0 \leftarrow x_2 \wedge x_3 \wedge x_7 \wedge \neg x_9$ | 1.4 |
| | $a_1 \leftarrow x_3 \wedge x_4 \wedge \neg x_5$ | 1.6 | | $a_1 \leftarrow x_3 \wedge \neg x_4$ | 1.6 |
| | $a_1 \leftarrow x_6 \wedge x_8 \wedge x_9$ | 1.4 | | $a_1 \leftarrow \neg x_6 \wedge x_8$ | 1.4 |
| | $a_2 \leftarrow \neg x_2 \wedge x_6 \wedge \neg x_7$ | 1.5 | | $a_2 \leftarrow \neg x_2 \wedge x_6 \wedge x_7 \wedge \neg x_8$ | 1.4 |
| | $a_2 \leftarrow x_5 \wedge \neg x_8 \wedge x_9$ | 1.6 | | $a_2 \leftarrow x_5 \wedge x_8$ | 1.5 |

**AI Agent Learning**  Our AI agent operates as a rule-based probabilistic decision-maker, initially configured with a subset of ground truth rules and some of which are wrong, as shown in Table. 4. This initialization mimics real-world scenarios where prior knowledge informs decision-making frameworks. Throughout the simulation, the AI agent dynamically updates its rule set and associated weights based on incoming data to refine its decision-making process.

Table 4: The rules assigned to a nascent AI agent.

| Head Predicate | Rules | Weight |
|---|---|---|
| $a_0$ | $x_0 \wedge x_1 \wedge \neg x_2$ | 1.2 |
| $a_1$ | $x_3 \wedge x_4 \wedge \neg x_5$ | 1.2 |
| $a_2$ | $x_2 \wedge x_6 \wedge x_7$ | 1.2 |

To facilitate the learning of these rules, our system relies on a reward feedback mechanism. Specifically, we simulate an oracle environment using the ground truth rule set. For each patient data instance, the AI agent proposes a treatment which is then evaluated by the oracle. The oracle assesses this treatment by computing the conditional probability of the treatment given the patient's data as depicted in Eq. 11. Subsequently, a reward is sampled using a Bernoulli distribution based on this probability. This reward signal serves as crucial feedback, enabling the AI agent to optimize its rule set for improved decision accuracy over time.

The metrics employed to evaluate the performance of rule learning are weight mean absolute error (MAE) and rule accuracy. The calculation of weight MAE follows a stringent methodology: for rules accurately identified in the ground-truth rule set, we directly compute the absolute error. For those not accurately identified, we assign the absolute error to be equal to the true weight. Regarding rule accuracy, if the rules in the ground-truth rule set are not accurately identified, we account for this in our evaluation. These metrics effectively assess the performance of our rule-learning method.

# E    EXPERIMENTAL RESULTS OF CUMULATIVE REWARDS

We employed a combination of methods across cumulative sample sizes, starting from 1,000 samples and incrementally increasing by 5,000 samples at each step, up to a maximum of 16,000 samples. The performance of these methods was evaluated using the same set of 100 evaluation samples across all stages. This approach ensures consistency in evaluation while progressively analyzing the impact of larger training datasets on the methods' performance.

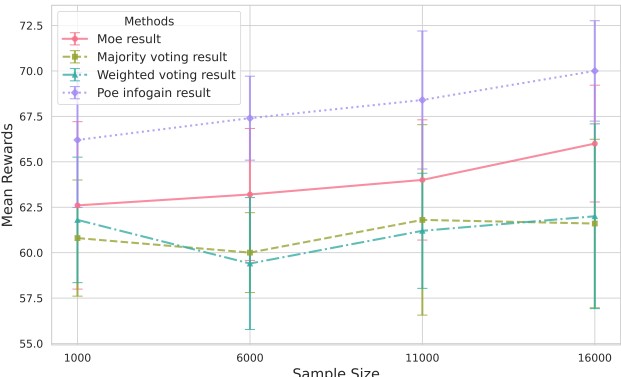

Figure 5: Temporal visualization of cumulative rewards.

# F    EXPERIMENTAL RESULTS OF RULE LEARNING

We report the rule learning accuracy of rule content and the MAE of rule weight in 5. Besides, we also report the agent's development by detailing the evolution of the rule set across varying training sample sizes in one of the repeated experiments, as shown in Table. 6.

Table 5: The rule learning accuracy of the AI agent across different sample sizes is illustrated here. Details on how to calculate rule accuracy and weight mean absolute error (MAE) can be found in Appendix D.

| Sample Size | Rule Accuracy | Weight MAE | Sample Size | Rule Accuracy | Weight MAE |
|---|---|---|---|---|---|
| 5000 | $0.55 \pm 0.08$ | $0.774 \pm 0.118$ | 10000 | $0.55 \pm 0.08$ | $0.757 \pm 0.117$ |
| 15000 | $0.80 \pm 0.07$ | $0.382 \pm 0.113$ | 20000 | $0.98 \pm 0.05$ | $0.134 \pm 0.065$ |

Table 6: Evolution of rule sets and weights possessed by the AI agent after processing every 5,000 samples. This table reports all learned rules during each update.

| Samples | Rule Sets | Weight | Samples | Rule Sets | Weight |
|---|---|---|---|---|---|
| 5000 | $a_0 \leftarrow x_0 \wedge x_1 \wedge \neg x_2$ | 1.5354 | 10000 | $a_0 \leftarrow x_0 \wedge x_1 \wedge \neg x_2$ | 1.5279 |
| | $a_0 \leftarrow x_3 \wedge x_7 \wedge \neg x_9$ | 1.3974 | | $a_0 \leftarrow x_3 \wedge x_7 \wedge \neg x_9$ | 1.4984 |
| | $a_1 \leftarrow x_3 \wedge x_4 \wedge \neg x_5$ | 1.2949 | | $a_1 \leftarrow x_3 \wedge x_4 \wedge \neg x_5$ | 1.3360 |
| | $a_2 \leftarrow \neg x_2 \wedge x_6 \wedge \neg x_7$ | 1.8813 | | $a_2 \leftarrow \neg x_2 \wedge x_6 \wedge \neg x_7$ | 2.0054 |
| 15000 | $a_0 \leftarrow x_0 \wedge x_1 \wedge \neg x_2$ | 1.5494 | 20000 | $a_0 \leftarrow x_0 \wedge x_1 \wedge \neg x_2$ | 1.5381 |
| | $a_0 \leftarrow x_3 \wedge x_7 \wedge \neg x_9$ | 1.4870 | | $a_0 \leftarrow x_3 \wedge x_7 \wedge \neg x_9$ | 1.4905 |
| | $a_1 \leftarrow x_3 \wedge x_4 \wedge \neg x_5$ | 1.5142 | | $a_1 \leftarrow x_3 \wedge x_4 \wedge \neg x_5$ | 1.5204 |
| | $a_1 \leftarrow x_6 \wedge x_8 \wedge \neg x_9$ | 1.5476 | | $a_1 \leftarrow x_6 \wedge x_8 \wedge \neg x_9$ | 1.5459 |
| | $a_2 \leftarrow \neg x_2 \wedge x_6 \wedge \neg x_7$ | 1.8329 | | $a_2 \leftarrow \neg x_2 \wedge x_6 \wedge \neg x_7$ | 1.5653 |
| | | | | $a_2 \leftarrow x_5 \wedge \neg x_8 \wedge x_9$ | 1.4854 |

## G  LIMITATION AND BROADER IMPACTS

A notable limitation of our proposed model lies in the rule-learning module of the branch-and-price-based column generation algorithm. While it effectively identifies rules that capture partial aspects of the ground truth, it occasionally fails to fully match clinical realities. Although stringent, these exact matches hold significant importance in clinical contexts where precision is paramount for diagnosis and treatment. In such scenarios, tailored strategies require more accurate rule-learning capabilities, highlighting the need for improvement in the robustness of our approach.

In real-world applications, the interaction between doctors and AI often occurs within a multi-agent system, where experts may take into account the perspectives of their peers and the AI's recommendations before reaching a final decision. This introduces a more interconnected and complex decision-making environment than our current framework, which treats each expert independently. Furthermore, while our framework supports human intervention to modify or remove high-risk rules, this manual process can be time-consuming and requires substantial expertise. Future work could focus on developing more user-friendly interfaces and automated tools to assist human experts in this task, potentially increasing both efficiency and adoption.

A promising direction for future research involves the introduction of hypernetworks to enable differentiable rule learning, which could improve the accuracy of rule discovery. However, purely data-driven approaches without expert knowledge may introduce noise and fail to capture patient-specific features. Therefore, combining knowledge-based and data-driven frameworks could enhance the robustness and accuracy of rule learning in clinical settings. Moreover, incorporating a human-in-the-loop algorithm would allow for the flexible integration of expert opinions in rule learning, further improving security and stability. While our column-generation algorithm produces relatively stable rules, these enhancements could better align the model with real-world complexities.

## H  COMPUTING INFRASTRUCTURE

All synthetic data experiments are performed on Ubuntu 20.04.3 LTS system with Intel(R) Xeon(R) Gold 6248R CPU @ 3.00GHz, 227 Gigabyte memory.

