# OpenReview forum: "Who Should Join the Decision-Making Table? Targeted Expert Selection for Enhanced Human-AI Collaboration"
_ICLR.cc/2025/Conference — Submitted to ICLR 2025_

### Official Review · Reviewer_HSvw · 2024-10-29

**Soundness:** 2
**Presentation:** 2
**Contribution:** 2
**Rating:** 5
**Confidence:** 4

**Summary:**

This paper presents a Product-of-Experts (PoE) model to enhance decision-making in AI-human collaborative systems, particularly in complex, high-stakes domains such as healthcare. The PoE model integrates a logic-informed AI agent with human expert opinions, leveraging probabilistic recommendations and expert reliability modelling to filter out noisy inputs and prioritize high-quality decisions. Notable features include targeted expert selection based on information gain and entropy reduction, the use of synthetic datasets for evaluation, and a structured approach to decision-making under varying levels of complexity. The model's primary goal is to improve collaborative efficiency by strategically selecting experts whose insights add the most value, thus reducing consultation costs and optimizing decision accuracy.

**Strengths:**

1) Writing is generally acceptable.
2) The sequential expert selection approach invites experts based on information gain, effectively filtering out less informative inputs. This could be critical in time-sensitive or resource-constrained environments like healthcare.
3) The model's use of confusion matrices to assess expert reliability provides a defence against unreliable or noisy inputs.
4) The incorporation of a logic-informed AI agent has the potential to make it interpretable by leveraging established medical knowledge and rules --- however, this has not been tested.

**Weaknesses:**

1) My biggest concern is with the evaluation. The evaluation lacks real-world case studies, which would provide valuable insight into how the model performs with actual patient data. This absence limits the credibility of the model's robustness. Specifically, the use of synthetic data and predefined rules provides controlled evaluation. However, it fails to convince me in terms of accounting for real-world complexities and noise in patient data and other contextual factors, which could unjustifiably reduce the perception of the expertise of doctors. This risks overestimating or underestimating the doctor's performance. The work remains untested in unpredictable scenarios.

2) I believe that the classification of doctors into narrow specialists, broader experts, and random decision-makers lacks the complexity of real-world expertise. This setup does not accurately reflect the variability and complexity of real-world expertise, potentially limiting its practicality to actual clinical settings.

3) The model's design favours convergence on a single recommended action, which may not reflect real-world cases where multiple treatment options could be equally viable. Yes, Level-1 scenarios kind of capture this, but the authors' intention for Level-1 scenarios was very different. This limitation affects cases where multiple and often complementary actions play a role.

4) How we can reliably compute the confusion matrices is unclear.

While the paper presents a promising framework, its oversimplified expert modelling, reliance on synthetic data, and lack of real-world validation reduce confidence in its readiness for real-world application.

**Questions:**

1. How would the model handle cases that require interdisciplinary collaboration across narrow specialties (e.g., when Doctors 1, 2, and 3 must work together)?

2) What are the authors' plans for validating the model on real-world patient data? Could they provide insights into how they might address the unpredictability and variance seen in genuine cases beyond synthetic data?

3) Following the previous question, what specific process was used to collect the domain knowledge? What were the specific sources consulted?

4) How would the model handle scenarios where multiple actions could lead to positive outcomes? Would the authors consider expanding the model to support probabilistic representation for multiple viable actions, allowing for flexibility in cases where a single action may not be optimal?

5) How do the authors envision healthcare professionals interacting with the model (and vice versa), particularly when interpreting the logic-informed AI recommendations alongside expert input? How could you convince the community that the proposed AI is the best way to select the set of doctors to work with the AI?

6) Given the reliance on historical reliability scores, how does the PoE model address potential bias against early-career doctors or those with limited track records? Have the authors considered adding mechanisms to ensure junior experts still have training opportunities or some input, particularly in non-critical cases?

---

> ### Author Response · Authors · 2024-11-25
> **Author response - Part 1 (Weaknesses)**
>
> > **Weakness 1 (The evaluation lacks real-world case studies)**:
>
> Thank you for your informative suggestion, and we deeply appreciate and agree with your concern.
> To address this, **we are actively working toward conducting additional experiments with actual patient data to strengthen the foundation of our work**. That said, we want to emphasize that our framework **was developed in collaboration with top experts specializing in rare diseases**. These experts have indicated that the framework is both practical and beneficial, as it reduces their workload while improving decision-making efficiency.
>
> Moreover, our simulation experiments, although based on synthetic data and predefined rules, have demonstrated that our method significantly outperforms other opinion integration approaches. While we acknowledge that synthetic evaluations may not fully capture the complexities and noise present in real-world patient data, the initial results, along with expert feedback, suggest the potential of our approach. We are committed to addressing this limitation through comprehensive real-world validation in future work, though we recognize that this process requires substantial resources and effort.
>
> > **Weakness 2 (Setup does not accurately reflect the variability and complexity of real-world expertise)**:
>
> Thank you for your insightful suggestion.
>
> To clarify, the simulation experiments in our manuscript were designed with the goal of **balancing interpretability and complexity**. While we acknowledge that real-world expertise is more nuanced and variable, our simplified framework serves as a starting point to demonstrate the model's potential. The classification into narrow specialists, broader experts, and random decision-makers allows us to systematically evaluate our framework's ability to handle different levels of expertise and variability in decision-making.
>
> We would like to emphasize that **increasing the number of doctors or further diversifying their profiles in the simulation is possible**; however, it would come at the expense of interpretability and controlled evaluation. Importantly, in real-world clinical scenarios, the key decision-making steps we aim to address often involve a limited set of critical choices, such as whether to recommend genetic testing for patients suspected of having Gitelman syndrome. Our framework is designed to focus on these pivotal moments, where expert opinions are essential and variability in expertise can significantly impact outcomes.
>
> Moving forward, we are committed to testing our framework in real-world settings to further confirm its applicability and robustness. We hope this explanation clarifies our intuition and reassures you about the value of the synthetic experiments in supporting our claims. Thank you again for your thoughtful feedback.
>
> > **Weakness 3 (How we can reliably compute the confusion matrices is unclear)**:
>
> Thank you for your interest in our work. The confusion matrices in our framework are **learned through a Maximum A Posteriori (MAP) estimation process, as detailed in Appendix B of the manuscript**. We also **conduct additional experiments to demonstrate the effectiveness of our estimation method in the response to the reviewer Ny3w**.
>
> In detail, this approach allows us to estimate the confusion matrices reliably, even in scenarios with limited data, which is a common challenge in real-world applications. To address this rare data challenge, we incorporate priors for each human expert's reliability. These priors can be adjusted based on prior knowledge of the expert’s performance or domain expertise, enabling the model to remain robust even when data is sparse.
>
> > **Weakness 4 (How would the model handle cases that require interdisciplinary collaboration across narrow specialties (e.g., when Doctors 1, 2, and 3 must work together?)**:
>
> Thank you for your interest in our work and for posing such an intriguing question.
>
> In the scenario you described, where interdisciplinary collaboration among Doctors 1, 2, and 3 is required, the cognitive framework of the human-AI collaboration would operate as designed to ensure effective integration of expertise. While it is rare for a single patient to satisfy all the rules of Doctors 1, 2, and 3 simultaneously, each doctor would make decisions independently based on their area of specialization.
>
> This scenario is conceptually similar to the challenge of handling narrow-AI models, where each AI specializes in a limited domain, as we discussed **in response to Reviewer sKnU (answer 4)**. **Even in cases where one or more decision-makers, including AI or humans, have limited or narrow expertise, our framework ensures robustness**. It does so by sequentially inviting experts based on estimated information gain, effectively managing the contributions of specialists without disproportionately amplifying the influence of a single narrow decision-maker.

---

> ### Author Response · Authors · 2024-11-25
> **Author response - Part 2(Questions 1-4)**
>
> > **Q1. What are the authors' plans for validating the model on real-world patient data?**
>
> Thank you for your helpful suggestion. **As mentioned in our response to Reviewer Ny3w, our next step is to validate the model on real-world patient data, focusing on cases of Cushing's syndrome and Gitelman syndrome**. We plan to collaborate closely with top experts in these areas to build a framework that assists in identifying patients who may benefit from genetic diagnosis.
>
> Cushing's syndrome often presents with symptoms that overlap with those of Gitelman syndrome or Bartter syndrome, which can make differential diagnosis challenging. By leveraging our framework, we aim to provide informative suggestions that guide experts in making more accurate decisions about which patients require further genetic testing. This approach not only supports clinical decision-making but also helps optimize resource allocation by reducing unnecessary tests.
>
> > **Q2. What specific process was used to collect the domain knowledge? What were the specific sources consulted?**
>
> Thank you for your interest in our work. To collect structured and compact domain knowledge, we follow a systematic process.
>
> 1. First, we collaborate with top experts in the field, leveraging their extensive experience and insights to define key decision points and diagnostic criteria.
> 2. Second, we consult authoritative sources such as medical handbooks, clinical guidelines, and peer-reviewed literature to ensure our framework aligns with established medical practices.
> 3. Lastly, we incorporate knowledge from existing resources like knowledge graphs or curated databases, which provide structured representations of domain-specific information.
>
> We appreciate your interest and are happy to provide further details if needed.
>
> > **Q3. How would the model handle scenarios where multiple actions could lead to positive outcomes?**
>
> Thank you for your thoughtful and insightful comment. We appreciate your concern and want to clarify that our model currently **provides a probability distribution as its output**, as **we have modify the Figure 1 to give a more vivid presentation**. This probabilistic output allows the final decision-maker, typically the responsible doctor, to evaluate multiple viable actions and choose the most appropriate course of action.
>
> When the probability distribution suggests that multiple actions have comparable probabilities of success, the decision-maker has the flexibility to pursue one or more actions as needed. This adaptability ensures that the framework can handle complex scenarios where a single optimal action may not exist.
>
> Additionally, because our decision-making process is designed to be interpretable and transparent, it enables the integration of other considerations such as patient preferences, available resources, or potential risks, enhancing the practical utility of the recommendations.
>
> We hope this clarification addresses your concern effectively, and we sincerely appreciate your feedback on this important aspect of our work.
>
> > **Q4. How could you convince the community that the proposed AI is the best way to select the set of doctors to work with the AI?**
>
> Thank you for your thoughtful comment. Our framework envisions healthcare professionals and the AI model working collaboratively, where the AI provides probabilistic, interpretable recommendations grounded in rule-based logic and continuously refined through patient feedback. Healthcare professionals retain ultimate decision-making authority, leveraging the AI’s probability distributions as a guide, particularly in complex or ambiguous cases.
>
> Our AI is designed to be transparent and interpretable, **allowing doctors to review the detailed knowledge-informed logic rules that influence the AI's recommendations**. Additionally, **doctors can examine the confusion matrices of each contributing expert to assess their reliability and determine whether to trust the AI's suggestions**. This transparency ensures that the AI serves as a trusted assistant rather than an opaque tool, fostering confidence and trust among medical professionals.
>
> Furthermore, as demonstrated through the theoretical analysis in Appendix C and our simulation experiments, our PoE model achieves near-optimal solutions efficiently, without excessive resource demands. By strategically selecting experts based on information gain, the framework ensures that only the most relevant inputs are incorporated, reducing inefficiency and enhancing decision accuracy. These features highlight the practicality and robustness of our approach, making a strong case for its effectiveness in selecting the appropriate set of doctors to collaborate with the AI.

---

> ### Author Response · Authors · 2024-11-25
> **Author response - Part 4 (Question 5)**
>
> > **Q5. Given the reliance on historical reliability scores, how does the PoE model address potential bias against early-career doctors or those with limited track records? Have the authors considered adding mechanisms to ensure junior experts still have training opportunities or some input, particularly in non-critical cases?**
>
> Thank you for your thoughtful suggestion. We appreciate and fully agree with your comment regarding the potential bias against early-career doctors or those with limited track records.
>
> Our human reliability model is designed using **Maximum A Posteriori (MAP) estimation**, which combines **prior knowledge** and **data-driven information** to address challenges associated with limited data. To mitigate potential bias, we can assign relatively lower priors to early-career doctors with limited track records, ensuring that their lack of historical data does not unfairly disadvantage them.
>
> Additionally, to provide training opportunities and encourage the involvement of junior experts with demonstrated potential, we can **adjust the priors to reflect their past performance or potential contribution**. For example, if a junior doctor has shown strong performance in previous cases, we could assign a higher prior to allow them more participation in non-critical cases, fostering their development and ensuring their input is considered.
>
> Thank you again for this important insight, and we will consider incorporating a discussion of this aspect into the manuscript to further enhance its comprehensiveness.

---

> > ### Comment · Reviewer_HSvw · 2024-11-26
> >
> > I thank the authors for their responses. I have reviewed them. While I acknowledge and appreciate the engagement with experts (which is really important in itself), the evaluation is still troubling due to the reasons I highlighted previously.

---

> > > ### Author Response · Authors · 2024-12-03
> > >
> > > Thank you for your valuable comments. We appreciate your concern about troubling evaluation. To address this, we have included a new semi-synthetic experiment that **integrates real-world healthcare data**, aiming to demonstrate the robustness of our method in real-world scenarios. We trust this additional experiment will clarify the issue, **and we have included the experiment setup and results in the global response**.

---

### Official Review · Reviewer_djtu · 2024-11-01

**Soundness:** 2
**Presentation:** 3
**Contribution:** 2
**Rating:** 5
**Confidence:** 4

**Summary:**

This paper proposed one AI-human collaborative framework with an active module which optimally select the human expert from the multi-disciplinary team. This framework includes the linear utility function, Plackett-Luce model, Product-of-Experts model, Human Expert Rilability model, and AI-Human ensemble model. The active learning idea is used to select the expert who can provide the most valuable contributions, such as the greatest informaiton gain or reduce the current entropy. In the model learning phase, MLE is used to learn the AI agent's logic rule and associated weights. MAP is used to estimate the parameters in the human's reliability model. The framework is tested based on the synthsized datasets with simulated patients and doctors.

**Strengths:**

- The idea, select the expert using the active learning algorithm, is good, which helps to save time and effort.
- Follow the existing Plackett-Luce model to model the probability distribution of the selected action.
- Fuse human opinions and AI recommendations via the product-of-experts model.

**Weaknesses:**

- The experiments do not fully support the advantages claimed for the proposed framework, specifically its robustness to influence from weak experts. Although the authors present some results in Appendix C, the robustness bound is unclear, and it is not evident how the advantage of the PoE model is demonstrated through the experimental results.
- The method for selecting the hyperparameter value is not clear.
- The proof explaining how targeted expert selection helps to find the optimal action is not well explained.

**Questions:**

- In the AI-human ensemble model, it is unclear how to set the value for the hyperparameter, \eta. This is particularly important if the authors aim to apply this framework in a healthcare setting, as small changes in \eta could significantly impact the final medical decision.
- In the targeted expert selection, why is the optimal solution guaranteed to be obtained by maximising information gain? How is the potential issue of minimising entropy leading to locally optimal solutions addressed?
- The experiments are not comprehensive. Although Mixture of Experts, majority voting, and weighted voting are included, these approaches are insufficient to fully evaluate the model.
- Why are level-0 and level-1 patients generated in this way? Does this imply that the proposed framework only works for the two-level scenario presented in the paper?

---

> ### Author Response · Authors · 2024-11-25
> **Author response - Part 1 (Weaknesses)**
>
> > **Weakness 1 (the advantage of the PoE model)**:
>
> Thank you for your detailed review and insightful comment. In our simulation experiments, we included **human doctors with varying levels of expertise across different cognitive regions**. This setup allowed us to compare each method's performance in evaluating and integrating expert knowledge. Compared to other methods, **the PoE model provides a more detailed modeling of each expert and integrates their decisions more effectively by combining the estimated distributions rather than the labels directly**. This approach is the most significant advantage of the PoE model.
>
> To further emphasize the advantage of the PoE model, we conducted **an additional experiment that included only the three best doctors from the simulation experiments**, all of whom specialized in at least two diseases. The results are presented in the following table. We observed that although the PoE model still performed the best, the gap between the PoE model and other methods was smaller. This finding indicates that the PoE model is particularly effective at combining detailed distributions, especially when integrating information from experts with varying levels of expertise, rather than simply aggregating labels.
>
>
> ### Level-0 Results
>
>
> | Methods                                 | Accuracy ↑         | Rewards ↑         | Regret ↓          |
> |-----------------------------------------|--------------------|--------------------|-------------------|
> | **AI agent**                            | 0.507 ± 0.035      | 66.4 ± 3.10        | 15.6 ± 6.55       |
> | **Doctor4**                             | 0.546 ± 0.053      | 67.3 ± 4.71        | 14.7 ± 7.14       |
> | **Doctor5**                             | 0.538 ± 0.040      | 67.3 ± 3.00        | 14.7 ± 7.13       |
> | **Doctor6**                             | 0.580 ± 0.032      | 68.3 ± 4.73        | 13.7 ± 5.88       |
> | **MoE**                                 | 0.402 ± 0.052      | 63.5 ± 4.57        | 18.5 ± 7.37       |
> | **Majority voting**                     | 0.385 ± 0.086      | 62.7 ± 6.21        | 19.3 ± 7.55       |
> | **Weighted voting**                     | 0.386 ± 0.084      | 61.3 ± 3.38        | 20.7 ± 5.40       |
> | **PoE + Infogain**                      | **0.532 ± 0.048**      | **65.9 ± 3.11**        | **16.1 ± 7.61**       |
> | **PoE + Infogain (no AI)**              | 0.391 ± 0.063      | 63.7 ± 5.69        | 18.3 ± 7.93       |
> | **PoE + Infogain (no AI calibration)**  | 0.523 ± 0.054      | 62.7 ± 3.80        | 19.3 ± 5.08       |
> | **PoE (greedy)**                        | 0.318 ± 0.038      | 61.1 ± 6.39        | 20.9 ± 10.68      |
> | **PoE (greedy) (no AI)**                | 0.333 ± 0.036      | 57.8 ± 4.77        | 24.2 ± 5.83       |
> | **PoE (greedy) (no AI calibration)**    | 0.328 ± 0.026      | 61.8 ± 5.15        | 20.2 ± 7.37       |
> | **GLAD**                                | 0.237 ± 0.017      | 56.7 ± 6.84        | 25.3 ± 7.52       |
>
> > **Weakness 2 (selection of hyperparameters)**:
>
> Thank you for your insightful comment and detailed review. We appreciate and understand your concern, and we will revise our manuscript accordingly to provide a clearer explanation. To clarify, our framework involves **only two sets of hyperparameters**.
> 1. The first is **the prior for the Maximum A Posteriori (MAP) estimation of each human reliability model**. In our experiments, we set the same prior for all participants, but in practice, higher priors can be assigned to top experts while relatively lower priors can be used for regular participants.
> 2. The **second hyperparameter is specific to the PoE model**. This includes defining **the upper bound threshold for the number of experts participating in the discussion and determining the convergence rate for the invitation process**. The process stops when the increase in information gain falls below this threshold. We appreciate your feedback and will ensure the manuscript includes these clarifications to improve its clarity. Thank you once again for your valuable comment.
>
> > **Weakness 3 (proof explaining how targeted expert selection helps to find the optimal action)**:
>
> We appreciate the reviewer’s observation regarding the explanation of how targeted expert selection aids in finding the optimal action. To clarify, our method is designed to approximate a near-optimal solution rather than guarantee the absolute optimal solution. The primary goal of our approach is to efficiently guide decision-making by leveraging the expertise of selected individuals.
>
> While we do not provide formal proof of optimality, our method consistently achieves near-optimal performance in empirical evaluations across various settings, as demonstrated in the experimental results (see Section 5.2). These results highlight the practical utility of our approach. We appreciate this constructive feedback.

---

> ### Author Response · Authors · 2024-11-25
> **Author response - Part 2 (Questions)**
>
> > **Q1. how to set the value for the hyperparameter $\eta$**:
>
> Thank you for your insightful comment. The calibration parameter, $\eta$, is indeed an important aspect of our framework. However, **it does not require manual definition**. As we mentioned in line 423, we employ **temperature scaling [1]** to optimize this parameter.
>
> Specifically, $\eta$ is calibrated with respect to the negative log-likelihood on the validation set, ensuring it is adapted to the data. This approach softens the softmax output of the AI without altering its maximum value. The primary purpose of this adjustment is to increase the entropy of the AI’s predictions initially, which reduces the AI’s influence during the early stages of the invitation process. This prevents overly confident AI outputs from prematurely dominating the decision-making process, thus ensuring a balanced integration of AI and human inputs. We will make sure to clarify this further in the revised manuscript. Thank you again for your valuable feedback.
>
> [1] Guo C, Pleiss G, Sun Y, et al. On calibration of modern neural networks[C]//International conference on machine learning. PMLR, 2017: 1321-1330.
>
> > **Q2. Why is the optimal solution guaranteed to be obtained by maximizing information gain? How is the potential issue of minimizing entropy leading to locally optimal solutions addressed?**:
>
> Thank you for your insightful and careful review. We apologize for the overstatement in line 83. As we have clarified in other parts of our manuscript, **our PoE model is designed to reach a near-optimal solution**, rather than guaranteeing the exact optimal solution. The framework maximizes information gain during expert selection to iteratively reduce uncertainty, which aligns well with minimizing entropy. However, we acknowledge that minimizing entropy can sometimes lead to locally optimal solutions.
>
> To mitigate this, our model incorporates a sequential invitation strategy, ensuring that the **selection process is adaptive and considers the marginal contribution of each expert's information**. This approach helps avoid getting stuck in local optima and improves the overall robustness of the solution. We appreciate your feedback and will revise the manuscript to better reflect these points and avoid overclaims. Thank you again for your valuable input.
>
> > **Q3. More comprehensive experiments**:
>
> Thank you for your insightful suggestion. As Reviewer b7JP also recommended, we have incorporated **an additional baseline method**, GLAD, into our experiments. The results, which include GLAD, can be found in **the response 2 to the Reviewer b7JP**. We would also like to emphasize **a key distinction between our method and other approaches such as MoE, GLAD, majority voting, and weighted voting**. Unlike these methods, which primarily focus on integrating all label-based or distribution-based decisions simultaneously, our method operates under a different premise. Specifically, at the outset, **we do not have access to the decisions of human experts**. Instead, our approach estimates the potential decisions and their associated reliability, then sequentially invites the most appropriate human expert based on this estimation. This sequential decision-making process sets our method apart from traditional ensemble approaches, which assume access to all opinions upfront for integration. We appreciate your suggestion and hope this clarification highlights the uniqueness of our framework.
>
>
> > **Q4. Why are level-0 and level-1 patients generated in this way? Does this imply that the proposed framework only works for the two-level scenario presented in the paper?**:
>
> Thank you for your detailed review, and we appreciate your concerns. We would like to clarify that **our framework is not limited to the two-level scenario presented in the paper**. The rationale for using Level-0 and Level-1 patients is grounded in real-world medical scenarios. In practice, some patients may present with only basic symptoms (Level-0), while a significant proportion may have additional complications and other symptoms (Level-1) that can influence the diagnosis and treatment decisions. This two-level structure allows us to model varying levels of complexity in patient presentations effectively. However, our framework is fully adaptable to scenarios involving more than two levels or different categorization schemes, depending on the application context. Thank you again for raising this important point.

---

### Official Review · Reviewer_sKnU · 2024-11-02

**Soundness:** 3
**Presentation:** 3
**Contribution:** 3
**Rating:** 6
**Confidence:** 3

**Summary:**

In this work, the authors propose a framework for improving human-algorithm decision-making by dynamically querying an ensemble of human experts. In particular, the authors develop a Product of Experts (PoE) model which factorizes the final probability of an action recommendation into a product of expert-specific and algorithmic decision recommendations. The authors also devise an entropy-based expert selection strategy that selects a subset of experts to inform decisions. The authors validate their framework theoretically and via synthetic experiments.

**Strengths:**

### Technical Framework

The authors propose a simple and interpretable approach for incorporating diverse expert feedback in collaborative algorithmic system recommendations. Both the rule-based controller and PoE are simple and easy to understand. I especially appreciated the
Advantages of PoE in Collective Intelligence subsection describing the rationale for PoE. This is quite interesting and I encourage the authors to add additional intuition for this point in the main paper. The targeted expert selection approach is also interesting and well-motivated.

### Motivation and Prior Work

The work has a clear grounding in prior work and demonstrates a novel set of approaches in comparison to existing approaches. The authors do a nice job of motivating the framework and providing a high-level overview early on in the work.

**Weaknesses:**

### Related Work:
- This paper generally covers relevant related work. There are several works that study expert heterogeneity in other contexts. For example, Rambachan (2024), Rambachan, Coston, & Kennedy (2024), and Lakkaraju et al. (2018) use heterogeneity as an instrumental variable for identification under unobservables. De-Arteaga et al. (2023) use heterogeneity to mitigate measurement error issues.

Rambachan 2024, Identifying Prediction Mistakes in Observational Data, https://academic.oup.com/qje/article-abstract/139/3/1665/7682113.

Rambachan, Coston, & Kennedy 2024, Robust Design and Evaluation of Predictive Algorithms under Unobserved Confounding, https://arxiv.org/abs/2212.09844.

De-Arteaga et al., Leveraging Expert Consistency to Improve Algorithmic Decision Support, https://arxiv.org/abs/2101.09648.

Lakkaraju et al. 2018, The Selective Labels Problem: Evaluating Algorithmic Predictions in the Presence of Unobservables, https://cs.stanford.edu/~jure/pubs/contraction-kdd17.pdf.

### Formal Model:
- The motivation for a simple rule-based agent is strong, as it offers transparency in high-stakes applications such as the running medical example. However, I do find it odd that algorithmic recommendations would be stochastic in this context (i.e., e.q 2). This would introduce additional variance in the selection of actions that may be unwanted by decision-makers. Similarly, there is a tension between the algorithmic recommendation being probabilistic and the human expert opinion being deterministic. (Line 244) Could you please elaborate on the rationale for this decision, and provide contexts in which it is likely to be reasonable? I understand this is necessary for the technical machinery but have concerns about the real-world applicability of this setup.
- The outputs of the AI-human ensemble model seem to imply that the algorithmic system will make the final decision (i.e., e.q. 6). However, I appreciated your clarification on lines 342-348 mentioning that humans have final authority over the decisions. This point would be helpful to make early on in the setup of the paper to help the reader understand the overall workflow.
- The indolence assumption imposed by the PoE model is strong. This assumes that human experts’ opinions do not influence one another. It would be helpful to provide a rationale for this assumption and a real-world context in which it is likely to be reasonable. Further, you might outline explicitly somewhere how you can get around this indolence assumption in the future. I understand that the assumption is helpful for the clean factorization and estimation approach, but would like additional justification.

### Experiments:
- Generally, the synthetic experiments demonstrate the utility of the proposed approach. I especially appreciated the ablations and Figure 2 illustrating the construction of the cognitive regions.
- However, I did find the setup of the synthetic experiment to be somewhat contrived. While the proposed design illustrates one configuration of regions in which the proposed approach improves accuracy and reward, I these results do not tell me how the proposed approach performs across an array of data generating processes (i.e., systematically varying the space of co-variates, decision rules, and other simulation parameters). Further, the experiments do not illustrate the specific failure modes of the proposed approach.
- Given that a key advantage of the proposed expert selection approach is information gain, it would be helpful to report the regret evaluated against the optimal configuration of experts for each decision instance. It would also be helpful to provide a temporal visualization of cumulative reward over time steps.
- This work would be strengthened substantively via an evaluation on real or semi-synthetic data. I understand that it is not possible to simulate environmental dynamics of collaborative decision-making in real time, but it would be helpful to illustrate how this framework might fit into a real-world decision-making workflow. Such an evaluation would also alay some of my modeling concerns (see above).

### Presentation:
- I find the semantics of “true action” (244) unintuitive. Is “true action” the final action that is proposed by the system?
- Minor: It is more standard (and readable, in my opinion) to omit boxes around equations. This is a minor point and I understand if the authors choose to keep the current format.

**Questions:**

- Could you describe a realistic real-world decision-making setting in which an algorithm would provide probabilistic predictions and human experts’ decisions would be deterministic?
- Under the simulation framework, could you elaborate on how the final decision of the collaborative team is generated at each time step? I checked through the appendix for the technical setup but this was not clear.
- Under what data generating conditions would the proposed PoE approach perform worse than MoE, majority voting, or other alternatives?

---

> ### Author Response · Authors · 2024-11-24
> **Author response - Part 1 (Formal model)**
>
> > **Weakness 1 (Questioning the randomness in algorithmic recommendations)**:
>
> Thank you for your insightful comment. In Equation 2, we employ the **Gumbel-Max trick**[1], which we'll explain in detail in our revised manuscript and provide proofs in the supplementary materials. The Gumbel-Max trick is commonly used to sample from categorical distributions by adding Gumbel noise to the logits and then selecting the maximum value. Prior research[2] has shown that the **Gumbel-Softmax technique allows for a smooth approximation of categorical distributions**.
>
> The rationale behind this approach is to enable the AI to provide a probabilistic distribution over possible decisions, capturing uncertainty inherent in complex medical scenarios. While human experts with different expertise typically offer deterministic, label-based decisions without detailed probability distributions. This approach is particularly reasonable in contexts where uncertainty is significant, such as diagnosing rare diseases or handling cases with limited data. It allows decision-makers to consider a range of possibilities informed by both AI and expert insights.
>
> [1]E. J. Gumbel, Statistical Theory of Extreme Values and Some Practical Applications: A Series of Lectures, Washington, D.C, USA:U.S. Dept. Commerce, vol. 33, 1954.
> [2]Jang E, Gu S, Poole B. Categorical reparameterization with gumbel-softmax[J]. arXiv preprint arXiv:1611.01144, 2016.
>
> > **Weakness 2 (humans have final authority over the decisions)**:
>
> Thank you for your interest in our work and for pointing out this important aspect. To clarify, **our intention is for the human decision-maker to retain final authority over all decisions**, which is crucial in high-stakes domains like healthcare to avoid medical liability and ensure ethical responsibility.
>
> In our framework, the AI system serves as an assistant rather than a final decision-maker. In Equation 6, we use the Product of Experts (PoE) model to integrate different opinions and apply a softmax function to produce a probabilistic distribution of the final decision. This distribution is then presented to the human decision-maker, who uses it to inform their judgment. We appreciate your suggestion and will **revise the manuscript to clarify this point earlier in the paper**.
>
> > **Weakness 3 (assumptions of the PoE model)**:
>
> Thank you for your insightful comment. We acknowledge that assuming independence among human experts in our PoE model is a strong assumption. While this simplifies our approach and makes estimation tractable, it is also reasonable in certain real-world contexts. For instance, in diagnosing rare diseases or in peer review processes, experts often provide independent judgments to prevent bias and undue influence.
>
> However, we understand that experts' opinions can sometimes influence one another. In future work, we plan to relax the independence assumption and develop models that account for dependencies among experts' opinions. This will allow us to capture a broader range of real-world interactions and enhance the applicability of our framework.

---

> ### Author Response · Authors · 2024-11-24
> **Author response - Part 2 (Experiments)**
>
> > **Weakness 1(Failure mode & data generating processes) **:
>
> Thank you for your interest and detailed review of our work. We appreciate and understand your concerns regarding the synthetic experiment setup.
>
> In our simulation experiments, our proposed approach consistently outperforms the baseline models. This is based on an important assumption: we have a generally reliable AI agent that possesses a rough understanding of the underlying patterns, even though it may still have uncertainties. It is crucial that this AI does not mislead the decision-making process—it should capture the main features and patterns without introducing significant errors.
>
> To address your concerns about the limited scope of our experiments and to better demonstrate the robustness of our approach under varying conditions, we conducted an additional simulation where the AI agent possesses only two of the six rules in the ground truth rule set, each assigned a random weight. This scenario simulates a less ideal situation where the AI's knowledge is incomplete. In this setting, while the PoE model does not exhibit the same level of superior performance as in the previous experiments, it still performs better than the other methods in two metrics.
>
> ### Level-0 Results
>
> | Methods                                 | Accuracy ↑         | Rewards ↑         | Regret ↓          |
> |-----------------------------------------|--------------------|--------------------|-------------------|
> | **MoE**                                 | 0.462 ± 0.037      | 66.8 ± 1.47        | 16.3 ± 2.97       |
> | **AI agent**                            | 0.449 ± 0.029      | 63.9 ± 4.13        | 19.2 ± 4.92       |
> | **Doctor1**                             | 0.376 ± 0.067      | 60.7 ± 4.29        | 22.4 ± 4.15       |
> | **Doctor2**                             | 0.437 ± 0.028      | 64.1 ± 5.37        | 19.0 ± 7.76       |
> | **Doctor3**                             | 0.395 ± 0.047      | 62.9 ± 4.66        | 20.2 ± 6.63       |
> | **Doctor4**                             | 0.557 ± 0.054      | 65.0 ± 4.29        | 18.1 ± 6.16       |
> | **Doctor5**                             | 0.563 ± 0.053      | 66.9 ± 5.09        | 16.2 ± 4.71       |
> | **Doctor6**                             | 0.575 ± 0.038      | 68.3 ± 4.29        | 14.8 ± 5.00       |
> | **Doctor7**                             | 0.300 ± 0.037      | 60.1 ± 3.96        | 23.0 ± 3.46       |
> | **Majority voting**                     | 0.331 ± 0.016      | 60.3 ± 4.08        | 22.8 ± 4.02       |
> | **Weighted voting**                     | 0.344 ± 0.034      | 61.2 ± 4.42        | 21.9 ± 5.52       |
> | **PoE + Infogain**                      | **0.519 ± 0.057**      | **65.6 ± 4.08**        | 18.0 ± 3.83       |
> | **PoE + Infogain (no AI)**              | 0.444 ± 0.039      | 63.7 ± 5.97        | 19.4 ± 5.73       |
> | **PoE + Infogain (no AI calibration)**  | 0.493 ± 0.043      | 65.1 ± 5.07        | **17.5 ± 5.51**       |
> | **PoE (greedy)**                        | 0.315 ± 0.026      | 59.6 ± 4.34        | 23.5 ± 5.32       |
> | **PoE (greedy) (no AI)**                | 0.331 ± 0.032      | 61.8 ± 4.51        | 21.3 ± 5.62       |
> | **PoE (greedy) (no AI calibration)**    | 0.325 ± 0.035      | 59.9 ± 2.30        | 23.2 ± 4.24       |
> | **GLAD**                                | 0.345 ± 0.035      | 61.3 ± 5.46        | 21.8 ± 7.56       |
>
>
> > **Weakness 2 (Regret & temporal visualization)**:
>
> Thank you for your insightful comment. To strengthen our analysis, we have included the regret metric to evaluate the performance of our method in both scenarios discussed in the manuscript. **These results have been detailed in our response to Reviewer b7JP**. As demonstrated, our PoE model consistently achieves the best performance. Additionally, **we have incorporated a temporal visualization of cumulative rewards over time steps in Appendix F**, providing a clearer view of the model's performance trends. We appreciate your feedback in helping us improve our work.
>
> > **Weakness 3 (real or semi-synthetic data)**:
>
> Thank you for your thoughtful feedback and for highlighting the importance of evaluating our framework on real or semi-synthetic data. We agree that such an evaluation would significantly strengthen our work by demonstrating the practical applicability of our approach in real-world settings. we are collaborating closely with clinical experts to map our framework onto existing clinical decision-making scenarios. In the revised manuscript, we plan to include a detailed case study that demonstrates the step-by-step integration of our framework within a real-world clinical workflow.

---

> ### Author Response · Authors · 2024-11-24
> **Author response - Part 3 (Presentation & Questions)**
>
> > **Weakness 1 (true action in line 244)**:
>
> Thank you for your detailed review and question. The term “true action” on line 244 refers to the **action actually implemented by the system**, based on specific features and parameters. This is distinct from proposed actions and is validated in Appendix B. We will clarify this term in the revised paper. Thank you for helping us improve the clarity of our manuscript.
>
> > **Weakness 2 (Equation box)**:
>
> Thank you for your helpful suggestion. We agree that omitting boxes around equations can enhance readability. We will revise our manuscript accordingly to make it clearer and more accessible to readers.
>
> > **Q1. a real-world decision-making situation where algorithms offer probabilistic predictions while human experts make deterministic decisions**:
>
> Thank you for your interest in our work. A practical example of a setting where an algorithm provides probabilistic predictions and human experts make deterministic decisions is in **medical diagnosis and treatment planning**. For instance, an AI model analyzing diagnostic images might estimate an 65% likelihood of Stage 1 cancer, a 30% chance of a benign tumor, and 5% uncertainty due to data limitations. Based on this, the oncologist must make a definitive decision, such as confirming the cancer diagnosis and recommending surgery followed by chemotherapy. AI’s probabilistic insights guide the expert’s decision, but it’s the human judgment that translates complex predictions into actionable steps, ensuring patient care is both precise and reliable.
>
> > **Q2. Could you elaborate on how the final decision of the collaborative team is generated at each time step?**:
>
> Thank you for your detailed review and interest in our work. We apologize for not clearly explaining how the final decision of the collaborative team is generated at each time step in our manuscript. We will **update this in our revised version.**
>
> In our simulation experiment, **to compare the performance of each baseline model, we generate the final decision by directly sampling from the final probability distribution produced by the PoE model**. This approach allows us to evaluate the expected performance by considering the stochastic nature of the decision-making process.
>
> However, **in a real-world implementation, the authority of the final decision is given to the human decision-maker who is responsible for the outcome**. The AI system provides probabilistic recommendations, but the final decision is made by the human.
> Thank you again for your helpful comment. We will ensure that our manuscript is updated accordingly to clarify this aspect.
>
> > **Q3. Failure mode**:
>
> Thank you for your insightful question. As we mentioned in our response to Part 2 - Weakness 1, our framework is built on the assumption of having a relatively well-performing AI agent. To address your concern, we conducted additional experiments to explore how the quality of the AI agent affects the performance of our PoE approach.

---

### Official Review · Reviewer_KtoA · 2024-11-03

**Soundness:** 2
**Presentation:** 3
**Contribution:** 2
**Rating:** 5
**Confidence:** 4

**Summary:**

The paper proposes a solution to the expert selection problem in human-AI collaboration. The AI in the paper is logic-based, which is interpretable by humans. The algorithm estimates the posterior recommendation of actions, conditioning on AI predictions and doctor decisions. The expert is then selected by maximizing the information gain. The method is evaluated on a synthetic dataset.

**Strengths:**

The question in the paper is novel and important. It proposes to select human experts into the decision-making team to maximize the marginal information gain. The approach improves the complementarity of the human-AI team.

**Weaknesses:**

I found the approach in the paper not convincing enough:

* Claim on interpretability: while the paper claims a lot about the interpretability and reliability of the system, I found few statements are evaluated in the evaluations. e.g.
  - the AI is constructed from logic rules, which the paper claims to be interpretable and transparent. However, compared to other black box learning algorithms, there are no evaluations of the interpretability or the reliability of such logic-rule-based AI predictions. It is unclear whether such a construction is necessarily good or how it affects the performance of the system.
  - the evaluations seem to be purely based on synthetic data. See my question below.

* Information gain: the paper takes an approach from rational decision-making by estimating the posterior distributions and maximizing information gain. While I agree that the posterior conditioning on a multi-dimensional signal space can be hard to estimate, I feel the PoE model is poorly justified. How well can it represent the posteriors and what are the alternatives? What are the assumptions? e.g. is it implicitly assuming the signals from the doctors are conditionally independent? Particularly, the following paper might be relevant:
> Ziyang Guo, Yifan Wu, Jason D. Hartline, and Jessica Hullman. 2024. A Decision-Theoretic Framework for Measuring AI Reliance. In Proceedings of the 2024 ACM Conference on Fairness, Accountability, and Transparency (FAccT '24).

* Greedy selection vs optimal: the algorithm for expert selection is a greedy algorithm that maximizes marginal information gain. However, there's no analysis of the algorithm being optimal or approximately optimal. In fact, this signal selection problem has been shown to be computationally hard:
> Chen, Y., & Waggoner, B. (2016, October). Informational substitutes. In 2016 IEEE 57th Annual Symposium on Foundations of Computer Science (FOCS) (pp. 239-247). IEEE.

**Questions:**

It seems like Table 1 compares the output of the PoE model with different setups. However, the paper claims: "In our AI-human collaborative system, it is important to note that the final decision rests with the human expert team, with the support of the AI agent in the decision-making process." I wonder how real human make decisions under a synthetic dataset, or the evaluation does not consider real human decisions.

---

> ### Author Response · Authors · 2024-11-24
> **Author response - Part 1**
>
> > **Weakness 1 (Claim on interpretability)**
>
> Thank you for your insightful comment. Our logic-rule-based AI is specifically designed to prioritize interpretability and transparency, especially in multidisciplinary team (MDT) settings, where trust and clear reasoning are critical. A significant bottleneck in MDT decision-making is the **"cognitive blind spots"** of experts—areas **where individual expertise may lack breadth or where decision-making can benefit from structured guidance**. Unlike black-box models, our approach uses logic rules, which **allows the decision-making process to be directly examined and verified by human experts, fostering collaboration and ensuring accountability**.
>
> In terms of reliability, the use of logic rules **inherently enforces constraints that reduce arbitrary or unpredictable behavior, ensuring the AI aligns with predefined safety protocols**. While we did not explicitly compare interpretability with black-box approaches in this paper, our design addresses the growing need for transparent AI, particularly in sensitive fields like healthcare.
>
> However, we acknowledge that further empirical studies are needed to systematically evaluate how these factors impact the system’s overall performance and usability. We plan to address this limitation in future work through targeted evaluations with both experts and comparative benchmarks.
>
> > **Weakness 2 (Information gain)**
>
> Thank you for your informative comment. We make two assumptions there:
>
> 1. The Al agent's recommendation $p^{\mathrm{AI}}(a \mid \boldsymbol{x})$ depends on the input feature vector $\boldsymbol{x}$.
>
> 2. The human expert $l$ 's opinion is deterministic for a given input $\boldsymbol{x}$, expressed as $h_l(\boldsymbol{x})=u_l$. The human model converts these deterministic outputs into probabilities $p_l\left(a \mid u_l\right)$, which depend only on the opinion $u_l$, not directly on $\boldsymbol{x}$.
>
> **We chose the PoE model because it offers a practical and computationally efficient way to combine multiple experts' opinions when estimating posterior distributions**. In our simulation experiments, the PoE model demonstrated superior performance, as reflected in our reward metrics, which implicitly **validate the quality of the posterior estimates**. Additionally, **as mentioned in our response to Reviewer Ny3w**, our method achieves stable and accurate estimations of each expert's reliability even with limited data.
> We recognize that the PoE model implicitly assumes that the signals from the doctors are conditionally independent given the true state. This means each doctor makes decisions independently without being influenced by others. While this assumption simplifies the modeling process, we agree that it may not always hold in real-world scenarios where experts could be influenced by shared information or common factors.
>
> As alternatives, models that account for dependencies among experts could be considered. However, these models often require more complex inference procedures and larger datasets. In future work, we plan to develop a more sophisticated framework that relaxes the conditional independence assumption.
>
> > **Weakness 3 (Greedy selection vs optimal):**
>
> Thank you for your thoughtful comment. We understand and appreciate your concern regarding the lack of analysis on the optimality of our greedy algorithm for expert selection.
>
> Our approach employs a greedy algorithm that maximizes marginal information gain. While we acknowledge that this method does not guarantee global optimality and that the signal selection problem is indeed computationally hard, we believe that **strict optimality is not essential for our framework to be effective in practice**.
>
> Empirically, **our experiments demonstrate that the greedy algorithm performs well**. For instance, in the visualization of the probability distribution during an invitation process (Figure 3), we observe that our solution can even surpass the ground truth in certain cases under the assumption of component independence. This suggests that the **greedy method achieves near-optimal results in practice**.
> The simplicity and efficiency of the greedy algorithm make it highly practical for real-world applications, where computational resources and time are often constrained. While theoretical guarantees of optimality are valuable, our empirical evidence shows that our method effectively balances performance and computational efficiency, making it suitable for practical deployment.

---

> ### Author Response · Authors · 2024-11-24
> **Author response - Part 2**
>
> > **Q1. The final decision-maker in our framework**
>
> Thank you for your insightful comments. We appreciate your perspective and the opportunity to clarify our approach.
>
> **Our research was initiated through discussions with clinical experts [1]**, aiming to develop a feasible framework that supports clinical decision-making while being acceptable to physicians in terms of computational demands and data requirements. Although we have not yet conducted evaluation experiments involving real human decisions, we believe that the accuracy of the AI system plays a crucial role in the final decision-making process, even when the ultimate decision rests with human experts.
>
> Clinical professionals have expressed that they value not only the accuracy of AI frameworks but also their practical feasibility and interpretability. Our proposed framework meets these requirements, which is not overly demanding on their workflow. In current hospital settings, similar AI assistance tools are already in use—for example, laboratory results often include AI-generated probability distributions to help clinicians assess the likelihood of conditions such as cancer.
>
> While we have not yet implemented our system in a hospital setting or evaluated it with real human decisions, our collaboration with clinical experts suggests that our approach aligns with their needs. Incorporating human decision-making into our evaluation is an important aspect of our future work. At this stage, our focus has been on demonstrating the potential of our framework using synthetic datasets to ensure its viability before clinical application.
>
> [1] Close M. 4.1 Artificial Intelligence (AI) Multi-Disciplinary Teams (MDTs) [Journal].

---

> ### Comment · Reviewer_KtoA · 2024-11-26
>
> Thank the authors for clarifying. This paper considers an important question, but I'm still concerned about synthetic data, especially when it seems generated from the same hypothesis as the model.

---

> ### Author Response · Authors · 2024-12-03
>
> Thank you for your detailed review. To address your concern regarding the use of only synthetic data in our experiments, we have conducted an additional semi-synthetic experiment **using real-world healthcare data in the context of the global response**. We hope this clarification helps to resolve your concern.

---

### Official Review · Reviewer_b7JP · 2024-11-03

**Soundness:** 3
**Presentation:** 3
**Contribution:** 2
**Rating:** 5
**Confidence:** 5

**Summary:**

This paper introduces a approach that leverages the Product of Experts (PoE) model to optimize decision-making by strategically
combining AI with human inputs. The proposed approach performed strategic selection of human experts based on how well their
knowledge complements or enhances the AI’s recommendations. Experiments in simulation environments demonstrate that the proposed method can effectively integrates logic rule-informed AI with human expertise, enhancing collaborative decision-making.

**Strengths:**

1. The research topic is timely and relevant. As human-AI collaboration becomes more prevalent, it is important to study how to effectively combine the strengths of both AI and human to achieve human-AI complementarity.

2. The paper is well-written overall and easy to follow.

**Weaknesses:**

1. The main contribution claimed by the authors is the algorithmic combination of AI probabilistic predictions with human deterministic decisions. However, this contribution closely mirrors the work presented in a NeurIPS 2021 paper [1], which similarly develops algorithms that integrate probabilistic model outputs with class-level human decisions. Equation (4) in the current paper is nearly identical to Equation (1) in the 2021 NeurIPS paper. Despite this resemblance, this foundational work is overlooked and is not included as an important baseline in either the discussion or experimental sections.

2. The evaluation has some limitations. 1) Limited experimental setting: The evaluation scenario is restricted to a specific medical decision-making task, leaving it unclear whether the proposed method generalizes to other AI-assisted decision-making contexts. 2) Limited baselines: In addition to the NeurIPS 2021 method, several established methods in collective intelligence and crowdsourcing, such as GLAD [2] and even the basic EM algorithm, are missing from the baselines. This absence raises questions about the method's effectiveness compared to well-established techniques.


3. Another important contribution of the paper claimed by the authors is that the approach can enhance model interpretability and human-AI interaction. However, there is no empirical support provided through human subject experiments to support these claims about the method's utility and effectiveness in ehancing human understanding of AI behavior or further improving the human-AI collarboration in decision making.

[1] Combining Human Predictions with Model Probabilities via Confusion Matrices and Calibration. Neurips 2021.

[2] Jacob Whitehill et al. “Whose vote should count more: Optimal integration of labels from labelers of unknown expertise”. In: Advances in neural information processing systems (2009).

**Questions:**

See above Weakness.

---

> ### Author Response · Authors · 2024-11-24
> **Author response - Part 1**
>
> > **Weakness 1 (Comparison to the NeurIPS 2021 Paper)**.
>
> Thank you for your thoughtful feedback. We greatly appreciate the NeurIPS 2021 paper, as it is an inspiring contribution to the field. However, we believe our approach addresses a different problem and focuses on a distinct use case.
>
> **1. AI's Role**: In contrast to their approach, where AI’s opinion is counted and integrated into the calibration process, our model positions AI as an organizer or inviter. AI in our setting selects and invites human experts based on their potential for information gain, but its own opinions are not considered in the final decision-making. The final decision rests solely with the human experts, aligning with the conventional  multi-disciplinary team (MDT) approach, where AI facilitates expert collaboration without directly influencing the outcome.
>
> **2. The Use of Confusion Matrix**: Their model uses the confusion matrix to combine and calibrate predictions from both AI and human inputs. In contrast, our approach takes an active perception strategy, using the confusion matrix to identify and invite the most reliable experts based on their expertise—without needing to know their actual opinions.
> We have already added the discussions to our revised draft.
>
> **3. Broader Scope of Collaboration**:
> Although both works utilize a confusion matrix to model human reliability, this is a common technique in the literature for combining multiple predictors [1], [2]. However, while the NeurIPS paper focuses on integrating the output of a single human decision-maker with the probabilistic model, our work expands this concept by enabling AI to collaborate with multiple human experts through product of expert model [3]. This broader collaboration allows for more dynamic decision-making across different contexts, which is a key distinction from the more limited scope of the NeurIPS model, which only integrates one human label-based predictor.
>
> We hope this clarifies the unique aspects of our approach. Thank you again for your valuable feedback. We would be happy to discuss any further questions you may have.
>
> [1]M. Steyvers, H. Tejeda, G. Kerrigan, P. Smyth, Bayesian modeling of human–AI complementarity, Proc. Natl. Acad. Sci. U.S.A.119 (11) e2111547119,https://doi.org/10.1073/pnas.2111547119 (2022).
>
> [2] Xu L, Krzyzak A, Suen C Y. Methods of combining multiple classifiers and their applications to handwriting recognition[J]. IEEE transactions on systems, man, and cybernetics, 1992, 22(3): 418-435.
>
> [3] Hinton G E. Training products of experts by minimizing contrastive divergence[J]. Neural computation, 2002, 14(8): 1771-1800.

---

> ### Author Response · Authors · 2024-11-24
> **Author response - Part 2**
>
> > **Weakness 2 (Experiments)**.
>
> Thank you for your thoughtful feedback and valuable suggestions. We appreciate the opportunity to address your concerns and further clarify our work.
>
> Unlike aggregation-based methods like GLAD or models that combine all opinions at once, **our approach focuses on selecting the next human decision-maker at each step of the process**. Importantly, at each stage of decision-making, we do not have access to the decisions made by future humans—this is a key distinction. Our method relies on selecting decision-makers based on maximizing information gain, without knowing the outcomes of subsequent decisions.
>
> To further address your concerns and demonstrate the robustness of our approach, we have conducted an additional experiment where we assume that all human decisions are available for all the other methods. In this experiment, our method still select the next decision-maker sequentially without their true labels, based on the objective of maximizing information gain. The results of this experiment are presented below, we will also update it in our revised paper
>
> ### Level-0
> | Methods                         | Accuracy ↑         | Rewards ↑         | Regret ↓          |
> |---------------------------------|--------------------|--------------------|-------------------|
> | AI agent                        | 0.502 ± 0.029      | 64.9 ± 3.91        | 15.3 ± 6.47       |
> | Doctor1                         | 0.405 ± 0.028      | 63.3 ± 4.65        | 16.9 ± 4.91       |
> | Doctor2                         | 0.452 ± 0.071      | 65.5 ± 3.56        | 14.7 ± 6.40       |
> | Doctor3                         | 0.403 ± 0.061      | 64.6 ± 2.69        | 15.6 ± 4.34       |
> | Doctor4                         | 0.551 ± 0.041      | 70.2 ± 3.25        | 10.0 ± 6.83       |
> | Doctor5                         | 0.553 ± 0.039      | 66.5 ± 5.12        | 13.7 ± 6.47       |
> | Doctor6                         | 0.561 ± 0.041      | 68.1 ± 3.75        | 12.1 ± 5.65       |
> | Doctor7                         | 0.341 ± 0.041      | 61.4 ± 3.14        | 18.8 ± 5.51       |
> | MoE                             | 0.486 ± 0.046      | 63.6 ± 5.75        | 16.6 ± 8.16       |
> | Majority voting                 | 0.332 ± 0.015      | 57.3 ± 3.49        | 22.9 ± 6.50       |
> | Weighted voting                 | 0.332 ± 0.015      | 59.9 ± 5.30        | 20.3 ± 6.18       |
> | PoE + Infogain                  | **0.588 ± 0.040**      | **67.4 ± 4.69**        | **11.8 ± 2.99**      |
> | PoE + Infogain (no AI)          | 0.433 ± 0.070      | 62.4 ± 6.36        | 17.8 ± 7.36       |
> | PoE + Infogain (no AI calibration)| 0.572 ± 0.044    | 68.4 ± 3.01        | 12.8 ± 4.14       |
> | PoE                             | 0.326 ± 0.031      | 60.7 ± 6.25        | 19.5 ± 8.51       |
> | PoE (no AI)                     | 0.330 ± 0.041      | 60.4 ± 4.34        | 19.8 ± 6.85       |
> | PoE (no AI calibration)         | 0.345 ± 0.038      | 62.4 ± 3.75        | 17.8 ± 4.66       |
> | GLAD                            | 0.334 ± 0.011      | 59.3 ± 4.90        | 20.9 ± 6.27       |
>
> > **Weakness 3 (human subject experiments)**.
>
> While we acknowledge that we have not yet conducted formal human subject experiments, our framework design has been developed **in close consultation with a team of doctors**, particularly those involved in multidisciplinary team (MDT) decision-making for rare diseases. These discussions highlighted **a critical bottleneck in MDT processes: cognitive blind spots** that can arise when handling complex cases.
>
> Our approach addresses this issue by using interpretable rules to provide insights that complement human expertise, ensuring the **AI does not disrupt or compromise the existing MDT workflow.** The current framework is designed to enhance collaboration **in a safe and feasible manner** by prioritizing alignment with current practices without introducing unnecessary risks.
> In future work, we plan to conduct empirical studies involving human subjects to provide stronger support for our claims and to further enhance the effectiveness of our approach in real-world settings.

---

> > ### Comment · Reviewer_b7JP · 2024-11-26
> >
> > Thanks for the authors' clarification. However, I am still concerned about the evaluation part of the paper, as it is conducted entirely on synthetic data without any real human participation.

---

> > > ### Author Response · Authors · 2024-12-03
> > >
> > > Thank you for your thorough review. Regarding your concern about the lack of real human participation, **we have conducted an additional semi-synthetic experiment using real-world healthcare data**, which we have posted in the **global response**. Specifically, this experiment **utilizes real-world data from rare disease patients**, incorporating more complex and realistic patterns. We trust that this will address your concern.

---

### Official Review · Reviewer_Ny3w · 2024-11-06

**Soundness:** 3
**Presentation:** 3
**Contribution:** 2
**Rating:** 6
**Confidence:** 2

**Summary:**

This paper leverages a Product of Experts (PoE) model that combines AI with input human expertise for enhanced decision-making, where an AI agent provides probabilistic, rule-based insights to choose a human expert for a specific case. The system dynamically selects human experts based on how well their expertise and recommendations might fit a specific scenario, while continuously updating its knowledge and selection criteria in an online setting. Through simulation experiments, the model demonstrates effective integration of rule-informed AI with human expertise.

**Strengths:**

This paper introduces a solution for a very relevant problem, and its a good idea for an AI to choose who to ask rather than making call themselves in critical settings like patient diagnosis.

**Weaknesses:**

1. I have a central problem with the premise of this paper.  The paper indirectly acknowledges an AI cannot be trusted to make direct medical decisions and it can be used well for augmenting human expertise. Yet it claims competence in selecting which human expert should make those decisions. This implies the system can reliably evaluate medical expertise, which might requires equal or greater medical knowledge than making the diagnoses itself. Maybe this is wrong, and a doctor's expertise pertaining to certain cases can be evaluated without any medica knowledge as its captures in the features $x$, but if the model is so good, why not use it for almost similar strategy for diagnosis. Instead of doctors, you choose diagnosis.


2. Almost all experiments are done on synthetic data, but many cases when it is not clear which expert to reach out to are the ones which are not so simple. And the rules for generating the synthetic data are fairly simple, so I am not sure how well it covers these rare boundary cases where such an AI system might actually be needed.

**Questions:**

1. Is it actually perception in active perception module?

2. So the AI cannot be trusted to make critical decisions, but can be trusted to choose the expert to make critical decisions? Why one failure mode is different than the other? What if a wrong expert is chosen?

3. Rule 2 in line 198 is actually not a rule but a guideline which is shown to be false. Sometimes a patient responds positively to a drug, but that’s because the drug is targeting a symptom not the actual cause of the symptoms. I understand this is an example, but it clearly a red flag in terms of author understanding of medical domains.

4. Who is providing these rules for feature construction? Are they given by experts and verified through a panel of experts?

5. In Equation 2, why $g_a$ is modeled as Gumbel noise? An explanation for this should be added in the paper.

6. Line 220: How would you recommend getting $w_a$ when due to less data MLE cannot be used confidently?

7. How much data will you need to build a Human Expert Reliability Model for each expert? And how do you plan to get this data in a real world setting?

8. Are you assuming that all experts are available all the time? In real world, an expert might not be available, and another expert whose score is less might be available and for a particular case might be able to give an equally good advice. I am guessing a naive approach will be to go to the next expert according to information gain. But then a lot of time can be wasted in asking the expert with highest information gain. How can we extend this model to incorporate such a scenario?

9. If actual medical data is used in the process, do you envision any privacy leakage concerns?

**Details Of Ethics Concerns:**

I am not sure if such an important approach relevant for a medical system can be tested solely on synthetic data to evalauate the correctness. If published, the paper can be cited and used in developing expert recommendation systems, and might work only for simple straightforward cases. Theoretically it all seems fine, but I do not have the expertise or ability to evaluate how it will perform on real-world data.

---

> ### Author Response · Authors · 2024-11-24
> **Author Response - Part 1**
>
> > **Weakness 1: the premise of the paper.**
>
> Thank you for your insightful question. We address your concern with the following points:
>
> **The Distinction Between Decision Support and Autonomous Diagnosis:**
>
> **1 Role of Expertise Evaluation**: The primary aim of our model is to identify the most suitable expert based on the specific characteristics of the case. This involves matching the case features with patterns of expertise demonstrated in historical data, which is distinct from having the detailed domain knowledge necessary for diagnosis.
>
> **2. Why It's Feasible**: Evaluating expertise does not require full diagnostic capabilities. Instead, it relies on identifying patterns of successful decision-making under similar conditions, which are captured in measurable metrics (e.g., past outcomes, specialty focus). This is fundamentally a data-driven matching problem, not a diagnostic problem. In our paper, we specifically employ confusion matrix elements derived from historical outcomes to assess each expert’s past performance
>
> **Why Not Use the Model Directly for Diagnosis?**
>
> **1. Al's Role in Diagnosis Is Different**: While the Al could theoretically contribute to diagnosis, its current design focuses on augmenting human decision-making by narrowing the choice to the most appropriate expert. Diagnosis requires a deeper, often case-specific understanding that involves synthesizing various sources of medical evidence in real time.
>
> **2. Ethical and Practical Barriers**: Deploying Al directly for diagnosis involves significant ethical, legal, and practical hurdles, including the need for interpretability, accountability, and trust. These challenges are mitigated when Al is used to assist human experts rather than replacing them.
>
> **Potential for Diagnosis Support**: While the primary goal of our framework is expertise matching, our AI model is also equipped with rule-based knowledge that can inform diagnostic processes. This knowledge serves two purposes:
>
> **1. Enhanced Matching Accuracy**: By integrating diagnostic rules, the system can better evaluate case complexities and match experts with cases requiring specific expertise.
>
> **2. Collaborative Insights**: The rule-based knowledge allows the AI to function as a collaborative tool, offering context-aware recommendations that complement expert judgment.
>
> Although the framework does not directly perform diagnoses, equipping it with diagnostic knowledge enriches its ability to evaluate expert decisions and contribute to refining diagnostic rules. This dual role reinforces the collaboration between AI and human professionals while ensuring a pragmatic and ethically sound approach to integrating Al into healthcare decision-making.
>
>
> > **Weakness 2: whether the simple rules used to generate it adequately capture complex boundary cases where the AI system would be most needed.**
>
> Thank you for your insightful comment. **Complexity in expert selection arises from cognitive blind spots, not disease complexity.** We would like to address your concern with the following points:
>
> **1. Clarification**: **Rare cases** or **boundary scenarios** are challenging not due to highly complex diseases, but because of **overlapping symptoms** or **misinterpretation of simple rules**. For example, Gitelman Syndrome can be misdiagnosed as hypokalemia caused by diuretics. However, the diagnostic rules for Gitelman Syndrome are straightforward (e.g., hypokalemia, low blood pressure, and low urinary calcium). We have provided the real rules below:
> * $ \text{Gitelman Syndrome} \leftarrow  \text{Hypokalemia} \land \text{Renal Potassium Wasting} \land \text{Normal or Low Blood Pressure} \land \text{Youth or Adult Onset} \land \text{Low Urinary Calcium} \land \text{No Response to Fludrocortisone Test}.$
> * $\text{Bartter Syndrome} \leftarrow\text{Hypokalemia} \land \text{Renal Potassium Wasting} \land \text{Normal or Low Blood Pressure} \land \text{Infant Onset} \land \text{Normal Urinary Calcium Levels} \land \text{No Response to Furosemide Test} $.
> * $\text{Cushing Syndrome} \leftarrow  \text{High Blood Pressure} \land \text{Low Renin} \land \text{High Aldosterone} \land \text{Multiple Symptoms including Skin Changes or Salt Retention}  $.
>
> **The challenge is recognizing when these simple rules should be applied**, which is often hindered by human cognitive blind spots or limited access to expertise.
>
> **2. Our AI’s role**: Our AI system is designed to detect when symptoms deviate from typical patterns or when a general practitioner should consult another doctor with specialized expertise.

---

> ### Author Response · Authors · 2024-11-24
> **Author Response - Part 2**
>
> > **Is it actually perception in active perception module?**
>
> We appreciate your interest in our active perception module. To clarify, **our active perception module is not focused on direct sensory perception as traditionally understood but is instead designed to optimize the selection of human experts.** The goal of this module is to identify and invite the most relevant experts based on how their knowledge complements the AI’s recommendations, thereby enriching the decision-making process with diverse perspectives. In our framework, this selective process does not require an assessment of the experts themselves; rather, it leverages historical data on each expert’s decision patterns to estimate their potential contributions. This allows our system to determine the "most informative" expert selections dynamically, using methods grounded in information gain and entropy minimization to guide expert choice and enhance collaborative decision-making [1].
>
> [1]. R. Bajcsy, "Active perception," in Proceedings of the IEEE, vol. 76, no. 8, pp. 966-1005, Aug. 1988, doi: 10.1109/5.5968.
>
> > **Why one failure mode is different than the other? What if a wrong expert is chosen?**
>
> Thank you for your insightful comment. We address your concerns as follows:
>
> **Why Expert Selection is Less Risky Than Critical Decisions:**
>
> Choosing the right expert is a **lower-risk process** than allowing the Al to make critical decisions. When the Al misidentifies an expert, the consequence is a potential misdirection to the wrong specialist. This mistake can still be **corrected by human review**, where the medical professional can reassess the situation and re-route the patient. In contrast, critical decisions such as diagnoses or treatments have **immediate, irreversible consequences** that are harder to reverse or correct once made.
>
>
> **Handling Misalignment in Expert Selection:**
>
> Although a wrong expert may be chosen, the Al facilitates **collaborative decision-making**, not independent action. Human professionals retain control, allowing them to reconsider and adjust decisions as needed. Additionally, the Al continuously improves its expert selection capabilities by learning from past cases, reducing the likelihood of future errors. Thus, expert selection is a safer, more flexible process, with the human experts in control, compared to the irreversible risks of having Al directly make critical medical decisions.
>
> > **Addressing Rule 2: "If a patient previously responded positively to treatment, continue with the same treatment."**
>
> **1. Justification of Rule $\mathbf{2}$ in Certain Scenarios:**
>
> In many medical situations, continuing with a treatment that has previously shown a positive response is a **sensible approach**, particularly in **chronic diseases** or conditions where the underlying cause is clear, and the symptoms are well-controlled with a particular treatment. For example, a patient with well-managed hypertension on a specific medication may continue the treatment regimen because it has been effective in controlling their symptoms and there is no indication of worsening or new conditions.
>
> **2. Probabilistic Nature of Our Model:**
>
> As described in our draft, our model is a rule-based **probabilistic** model, where the **rules weights** and even the rules are not fixed but **are personalized based on the patient's individual history and current condition**. For example, Rule 2—"If a patient previously responded positively to treatment, continue with the same treatment"—**is not applied deterministically**. Instead, the weight of this rule is adjusted dynamically based on ongoing data, such as the patient’s evolving response to treatment and additional contextual factors. As new information becomes available, the model recalculates the probability of continuing the same treatment versus suggesting alternatives, allowing for flexible and personalized decision-making. In this way, **each rule is not a rigid directive but a probabilistic recommendation that adapts to the patient's specific situation.**
>
> > **Who is providing these rules for feature construction? Are they given by experts and verified through a panel of experts?**
>
> Thank you for your insightful question. While these rules are initially derived from a wide range of sources, such as expert input, medical knowledge graphs, consensus documents, and reputable medical handbooks,  the final verification is done by a panel of experts to ensure clinical accuracy and safety.

---

> ### Author Response · Authors · 2024-11-24
> **Author Response - Part 3**
>
> > **In Equation 2, why $g_a$ is modeled as Gumbel noise? An explanation for this should be added in the paper.**
>
> Thanks for your interest in the **Gumbel-max trick [1]**, and your helpful suggestion, we will add the explanation of this trick in our manuscript, and provide proof in the supplementary materials. As for the Gumbel Max trick, it's a method often used to sample from categorical distributions. Essentially, we add Gumbel noise to the logits, and then select the maximum value. The former research has proved that Gumbel-softmax can smoothly annealed into a categorical distribution[2].
>
> [1]E. J. Gumbel, Statistical Theory of Extreme Values and Some Practical Applications: A Series of Lectures, Washington, D.C, USA:U.S. Dept. Commerce, vol. 33, 1954.
>
> [2]Jang E, Gu S, Poole B. Categorical reparameterization with gumbel-softmax[J]. arXiv preprint arXiv:1611.01144, 2016.
>
> > **Line 220: How would you recommend getting $w_a$ when due to less data MLE cannot be used confidently?**
>
> Thank you for your valuable feedback. Given the challenge of insufficient data for MLE, we recommend using Maximum A Posteriori (MAP) estimation, with prior weights $w_a$ provided by domain experts. These relative weights determine the importance of each rule in decision-making and can be refined as more data becomes available. The parameter $ w_a $ represents the weight of each rule, and it reflects the relative importance of each rule in the decision-making process. A higher value of $ w_a $ means that the model will give more attention to that particular rule. Since $ w_a $ is a relative measure, it does not require precise estimation but rather indicates the importance of one rule compared to others.
>
> > **How much data will you need to build a Human Expert Reliability Model for each expert? And how do you plan to get this data in a real world setting?**
>
> To build a Human Expert Reliability Model, we estimate the confusion matrix based on **historical doctor assessments** and **patient outcomes**. In a real-world setting, we will collect data from electronic health records (EHRs), patient feedback, and expert assessments for continuous improvement. As proposed in our draft, we use MAP estimation to address sparse data challenges, incorporating priors and refining the model as more data becomes available.
>
> **Furthermore, to demonstrate the effectiveness of our MAP method in building a human expert reliability model for each expert, we conducted a simulation experiment.** We computed the RMSE between the estimated confusion matrix and the actual confusion matrix of each expert using 10,000 samples. We observed that when the sample size exceeds 1,000, the RMSE nearly converges, indicating that our model stabilizes with sufficient data.
>
> | Sample Size | Mean RMSE | Standard Deviation |
> |-------------|-----------|--------------------|
> | 200         | 0.265362  | 0.021634           |
> | 400         | 0.258338  | 0.013992           |
> | 600         | 0.246361  | 0.012967           |
> | 800         | 0.247735  | 0.011666           |
> | 1000        | 0.243126  | 0.010767           |
> | 1200        | 0.240466  | 0.010121           |
> | 1400        | 0.242673  | 0.008512           |
> | 1600        | 0.242171  | 0.006798           |
> | 1800        | 0.243299  | 0.008101           |
> | 2000        | 0.239353  | 0.007300           |
>
> > **Are you assuming that all experts are available all the time?**
>
> Thank you for your valuable feedback. We understand your concern about the availability of experts in real-world scenarios.
>
> This is actually **one of the key advantages of our framework when applied in practice**. Unlike traditional multidisciplinary team meetings, where all experts must be available at the same time and gather in one meeting room, **our online system is designed to assign specific tasks to the most appropriate expert based on the situation**. This allows experts to **review patient records and provide their input remotely in their convenient time**, without the need for all experts to gather in a single meeting room.
>
> In your mentioned case, our system can prioritize experts dynamically, minimizing time spent waiting for the highest-information expert and allowing for more efficient decision-making.
>
> > **If actual medical data is used in the process, do you envision any privacy leakage concerns?**
>
> Thank you for your comprehensive feedback.
> 1. We will prioritize data privacy by **applying anonymization and de-identification** techniques to remove personally identifiable information before using any data in the system.
> 2. Our rule-based model and confusion matrix estimation can be established **using only domain knowledge and doctor’s performance data (doctor’s assessment and patient outcome)**, without requiring direct access to sensitive patient information.
> 3. **The final decision-making authority rests with human doctors**, ensuring that sensitive patient data is handled only by authorized professionals throughout the process

---

> > ### Comment · Reviewer_Ny3w · 2024-11-25
> >
> > Thank you for the detailed response. I still believe that this work is important, and the only major change I would've loved to see is the evaluation of real data. I will increase the score to reflect this update.

---

> ### Author Response · Authors · 2024-11-26
> **Thanks for your interest!**
>
> Thank you for your valuable feedback and interest in our work. We really appreciate your expectations about evaluating real-world data. We are currently working with top healthcare experts in the region to assess and implement our framework.
> However, working with real-world medical data presents several challenges. It requires extensive data cleaning to address issues like missing or inconsistent information and strict measures to protect privacy. These tasks take time, and because of the complex regulatory and ethical standards involved, it’s difficult to predict an exact timeline. We are committed to managing these processes carefully to ensure both data quality and privacy, but the steps involved are often more time-consuming than anticipated.

---

### Author Response · Authors · 2024-12-03
**Additional semi-synthetic experiment**

Dear Chairs and Reviewers,

We sincerely thank all the reviewers for their time and valuable feedback, which has significantly contributed to improving our work.

In response to the reviewers' suggestions, **we have conducted additional experiments using real rare disease patient data (Gitelman syndrome) to further validate our approach**. Gitelman syndrome is a rare hereditary tubular kidney disorder that is difficult to distinguish from Bartter Syndrome and Cushing Syndrome. Experts in rare diseases typically recommend genetic testing to ensure accurate diagnosis. Therefore, our goal is to integrate the expertise of both AI and human doctors to make precise diagnostic decisions. Below are the details of the dataset and the experimental setup:

### **Dataset**

- **Gitelman Patients**: 71
- **Non-Gitelman Patients**: 95

Each patient record includes the following five features, which are critical for diagnosing Gitelman syndrome in clinical practice:
1. **Serum Potassium**
2. **Urine Potassium**
3. **pH**
4. **Bicarbonate**
5. **High Blood Pressure**

Additionally, each patient is labeled based on whether they have been diagnosed with Gitelman syndrome.

### **Experimental Setup**

We employed a **cross-validation approach**, where **one fold is used as the evaluation dataset and the remaining folds serve as the training set**. In the training set, our synthetic human doctors make decisions, and we use weighted voting to determine the final decision for each sample. If the decision matches the true label, a reward of 1 is given; otherwise, the reward is 0. We then estimate the reliability of the human model from the training set and evaluate the model's performance on the evaluation dataset. Our decision-making framework incorporates both human doctors and AI agents, with logic-informed mechanisms to effectively control the experimental scenarios. The specifics of our setup are as follows:
- **Participants**:
  - 6 Human Doctors:
    - 4 doctors (doctor 1, 2, 3, 5) operate within their unique cognitive regions, each utilizing a subset or a modified version of the overall rule set.
    - 1 doctor (doctor 6) employs a nearly random decision-making strategy.
    - 1 doctor (doctor 4) utilizes the complete overall rule set without modifications..

  - 1 AI agent:
    - Utilize 3 rules of the overall rule set.

- **Rule Sets**:
  The complete rule set used in the experiments is provided below. Each human doctor's rule set is derived from this comprehensive set, with subtle modifications to individual rules as necessary.

```plaintext
Gitelman:
  - pH > 7.45 ∧ serum potassium < 3.0 ∧ urine potassium > 20 ∧ bicarbonate > 24 ∧ high blood pressure = 0
    Weight: 2.5
  - pH > 7.45 ∧ serum potassium < 3.5 ∧ urine potassium > 25 ∧ bicarbonate > 24 ∧ high blood pressure = 0
    Weight: 2.5

Non-Gitelman:
  - pH < 7.35 ∧ serum potassium < 3.5
    Weight: 1.5
  - high blood pressure = 0 ∧ pH < 7.35
    Weight: 1.3
  - high blood pressure = 1 ∧ pH > 7.45
    Weight: 1.3
  - high blood pressure = 0 ∧ urine potassium < 20 ∧ pH > 7.45 ∧ bicarbonate < 22
    Weight: 1.5
```
### **Results**

We report the **mean accuracy across the cross-validation for each method**, as we do not possess a ground truth dataset that captures all patterns in these patients. Therefore, computing regret and reward metrics is not feasible. The results are summarized in the table below:

| Methods                                 | Accuracy ↑         |
|-----------------------------------------|--------------------|
| **AI agent**                            | 0.655 ± 0.033      |
| **Doctor1**                             | 0.772 ± 0.059      |
| **Doctor2**                             | 0.634 ± 0.017      |
| **Doctor3**                             | 0.681 ± 0.059      |
| **Doctor4**                             | 0.800 ± 0.056      |
| **Doctor5**                             | 0.703 ± 0.056      |
| **Doctor6**                             | 0.580 ± 0.027      |
| **MoE**                                 | 0.710 ± 0.080      |
| **Majority voting**                     | 0.675 ± 0.083      |
| **Weighted voting**                     | 0.686 ± 0.084      |
| **PoE + Infogain**                      | **0.765 ± 0.085**      |
| **PoE + Infogain (no AI)**              | 0.655 ± 0.093      |
| **PoE + Infogain (no AI calibration)**  | 0.675 ± 0.094      |
| **PoE (greedy)**                        | 0.537 ± 0.129      |
| **PoE (greedy) (no AI)**                | 0.517 ± 0.126      |
| **PoE (greedy) (no AI calibration)**    | 0.531 ± 0.126      |
| **GLAD**                                | 0.707 ± 0.083      |



We appreciate the reviewers' constructive comments and believe that the additional experiments and clarifications have strengthened our manuscript. **We will include these updated experiments in the final version of our manuscript**.

Thank you for your consideration.

Kind regards,

The authors

---

### Meta-Review · Area_Chair_wiUR · 2024-12-19

**Metareview:**

The reviewers acknowledged that the paper tackles an important problem of optimizing the selection of experts in a human-AI collaborative decision-making system, and the proposed approach could be potentially impactful. However, the reviewers pointed out several weaknesses and shared concerns related to the lack of experiments in real-world settings, limited empirical evidence supporting the paper's contributions, and limited engagement with closely related prior work. We want to thank the authors for their detailed responses. Based on the raised concerns and follow-up discussions, unfortunately, the final decision is a rejection. Nevertheless, this is exciting and potentially impactful work, and we encourage the authors to incorporate the reviewers' feedback when preparing a future revision of the paper.

**Additional Comments On Reviewer Discussion:**

The reviewers pointed out several weaknesses and shared concerns related to the lack of experiments in real-world settings, limited empirical evidence supporting the paper's contributions, and limited engagement with closely related prior work. A majority of the reviewers support a rejection decision and agree that the paper is not yet ready for acceptance.

---

### Decision · Program_Chairs · 2025-01-22

Reject